# In situ synthesis and dynamic simulation of molecularly imprinted polymeric nanoparticles on a micro-reactor system

Özgecan Erdem[1], Ismail Eş[1], Yeşeren Saylan[2], Maryam Atabay[1,2], Murat Alp Gungen[1,3], Kadriye Ölmez [1,3], Adil Denizli[2] & Fatih Inci [1,3] ✉

Current practices in synthesizing molecularly imprinted polymers face challenges—lengthy process, low-productivity, the need for expensive and sophisticated equipment, and they cannot be controlled in situ synthesis. Herein, we present a micro-reactor for in situ and continuously synthesizing trillions of molecularly imprinted polymeric nanoparticles that contain molecular fingerprints of bovine serum albumin in a short period of time (5-30 min). Initially, we performed COMSOL simulation to analyze mixing efficiency with altering flow rates, and experimentally validated the platform for synthesizing nanoparticles with sizes ranging from 52-106 nm. Molecular interactions between monomers and protein were also examined by molecular docking and dynamics simulations. Afterwards, we benchmarked the micro-reactor parameters through dispersity and concentration of molecularly imprinted polymers using principal component analysis. Sensing assets of molecularly imprinted polymers were examined on a metamaterial sensor, resulting in 81% of precision with high selectivity (4.5 times), and three cycles of consecutive use. Overall, our micro-reactor stood out for its high productivity (48-288 times improvement in assay-time and 2 times improvement in reagent volume), enabling to produce 1.4-1.5 times more MIPs at one-single step, and continuous production compared to conventional strategy.

Molecularly imprinted polymers (MIPs) are considered synthetic antibody- or enzyme-like receptors consisting of specific binding sites for target analytes[1]. They therefore have garnered notable attention as promising alternatives to biological receptors in multifaceted areas such as chemical sensing and biosensing[2,3], separation and purification[4,5], enzyme-like catalysis[6], drug development[7], and numerous biomedical applications[8–10]. MIPs denote highly selective binding sites towards their imprinted target molecule. Since they are synthesized as highly cross-linked polymeric structures, they are robust and stable, offering resistance to many extreme conditions such as acidic, basic, and high-temperature milieus. Furthermore, the fact that MIPs can be stored at room temperature makes them more advantageous compared to antibodies[11]. Owing to all these advances, MIPs have been employed in a myriad of sensing applications in medicine, security, and environmental monitoring through their integrations with optical, electrochemical, and photoelectrochemical sensing[12–15]. As an example, an electrochemical sensor translates interactions between an analyte and a receptor accommodating on the surface of an electrode into a quantifiable output signal (e.g., potential, current, conductivity or impedance)[16]. Viral proteins[17–19] and protein biomarkers[20–22] were successfully detected on MIPs-incorporated electrochemical sensors. However, varying ionic strength and content in biospecimens hampers their performance. Hence, washing out the sensors with non-ionic solutions would be a necessary step to

[1]UNAM-National Nanotechnology Research Center, Bilkent University, 06800 Ankara, Turkey. [2]Department of Chemistry, Hacettepe University, 06800 Ankara, Turkey. [3]Institute of Materials Science and Nanotechnology, Bilkent University, 06800 Ankara, Turkey. ✉e-mail: finci@bilkent.edu.tr

improve the signals[23], and also, the use of such materials with certain electrochemical properties would differentiate the signals from the artifacts. In particularly, conducting polymers (e.g., polyaniline, poly(3,4-ethylenedioxythiophene), polypyrrole, and polythiophene) exhibit notable electrical capacitance[24–27] or transfer electrical charges from some redox proteins and other biological entities[28]. They can be implemented onto the surface of electrodes, thereby forming mechanically stable layers via chemical synthesis[29], electrochemical deposition[24], enzymatic formation[30], and microorganism-assisted polymerization[31–33]. Among these conducting polymers, polypyrrole has been utilized in forming MIPs for detecting low number of molecules such as proteins[34,35], antibiotics[36,37], ions[38,39], and so on. On the other hand, optical sensors employ light-matter coupling scheme to convert binding actions into quantifiable signals[40], and this strategy has been applied into the detection of proteins, viruses, and extracellular vesicles[41–44]. Yet, the refractive index of the medium mostly dominates the optical signals while applying biospecimens. Thereby, refreshing the medium after the binding needs to be sufficient, and also, integrating anti-fouling agents would be a key step for minimizing such interfering factors.

Considering analytes, MIPs are very versatile that can be prepared in accordance with plentiful targets, and hence, they can be used for detecting analytes, such as viruses[45], bacteria[46], various biomarkers[47–49], proteins[50,51], and chemical contaminants[52,53]. In particular, proteins are pivotal disease biomarkers and essential macromolecules of organisms that involve in inter- and intracellular activities[54–57]. MIPs have been designed in detecting protein biomarkers because of their relevance in health-related processes[58]. In this manner, much emphasis has been given to the preparation of polymers for precise and selective isolation of proteins from complex biospecimens, particularly for biomedical and diagnostic applications[59]. In contrast to smaller templates, proteins are complex biopolymers with a wide variety of functional groups accessible for interacting with functional monomers. Their varying regions would have significantly distinct physicochemical characteristics, such as hydrophilicity/hydrophobicity, molecular grooves, and different charges[60]. Additionally, a number of protein imprinting techniques, such as boronate affinity-based molecular imprinting[61], solid-phase synthesis[62], and post-imprinting modification[63], have also been developed for the applications of separation[64] and purification[65], proteomics, biomarker detection[48], bioimaging[66], and therapy[67].

Basic strategy in fabricating MIPs includes interacting a template molecule, cell or compound with a suitable functional monomer, generating a pre-complex. In the presence of a cross-linker, initiator and pre-complex, a template-imprinted polymeric matrix is formed. After the removal of template with suitable desorption agents, a polymer with specific cavities is molded for recognition, adsorption, separation, and sensing applications[68]. The most prevalent techniques employed in MIPs synthesis include bulk[69–71], precipitation[72,73], emulsion[74,75], suspension[76,77], core–shell polymerization[78,79], and electrochemical methods[80]. In current practice, MIP production is employed as a bulk process, and basically, their synthesis is accomplished by magnetically stirring or shaking polymeric mixture in a closed environment. Such techniques are usually time-consuming (~24 h) and labor-intensive (multiple steps up to 10), and also require sophisticated equipment impeding their productivity. Continuous synthesis of MIPs is highly challenging and possesses severe limitations in large-scale production[81]. In particular, size variations, defects, and stability issues in different batches need to be addressed critically[82]. Otherwise, these issues severely impact reproducibility, thereby hindering the translations of MIPs into the market. In particular, scaling-up the synthesis of MIP nanoparticles faces many bottlenecks since the increased volume of reagents and dispersion volumes requires precise control of the batch for uniform mixing of the reagents to form reproducible MIP nanoparticles. Minute perturbations in the experimental conditions would impede the polymerization and growth of MIP nanoparticles, eventually leading to low-yield and polydisperse particle production[83]. This issue also points out the need for in situ monitoring of the nanoparticles for their manufacturing through a feedback loop mechanism, thereby assisting in effective decision-making processes[84]. Consequently, new platforms and strategies need to be deployed for accelerating the MIPs production process, at the same time for keeping the reproducibility and high-yield as high as possible, and reducing the overall cost.

On the other hand, microfluidics has emerged as an innovative technological tool and has been widely employed in nanoparticle synthesis for decades[85]. It has great advantages over conventional methods, such as low consumption of reagents, precise experimental control, reduced turnaround time, low energy consumption, maximum yield capacity, facile automation, continuous production/reaction, and efficient heat and mass transfer, allowing a more predictable scale-up strategy[86–90]. Principally, microfluidics involves the flow of complex fluids in mono- or multi-phase in artificial microsystems designed through different microfabrication techniques. Among different methods, stereolithography techniques allow rapid (only a few hours) and facile fabrication (single step) of microfluidic devices with desired geometry for nanoparticle synthesis. To date, a limited number of microfluidic approaches has been implemented for MIPs synthesis; however, they are not able to produce imprinted nanoparticles down to nm-scale[91,92]. Although recent studies incorporating a long tubing design that requires lengthy time for mixing the reagents have addressed the size-related challenges by synthesizing them down to ~250 nm, they are prone to produce nanoparticles with very low-yield (up to 30%); the variations in tubing design, total flow rate; and flow rate ratio would highly hinder their reproducibility; and their synthesis requires for 24 h, impeding to scale-up the production[81,93]. Hence, microfluidic-stemmed platforms need to be revitalized through comprehensive in situ analyses in order to produce imprinted nanoparticles within nm-scale, thereby boosting the surface-to-volume ratio that allows to interact more molecules, at the same time presenting high-yield, short production time, affordability, and facile-use assets.

Herein, we present a micro-reactor to in situ synthesize protein-imprinted nanoparticles through a continuous flow strategy. This system was initially strategized through COMSOL Multiphysics simulations to analyze mixing efficiency with different flow rates for mixtures and experimentally validated for synthesizing nanoparticles down to nm-scales (Fig. 1). In addition, molecular docking method was employed to predict the preferred orientations when a ligand and MIPs were reacted to form a stable complex. The micro-reactor led to the synthesis of MIPs in a relatively short time (down to 5–30 min) compared to conventional bulk production (24 h) for synthesizing the intended size of nanoparticles. As a model system, bovine serum albumin (BSA) was interacted with methacrylic acid as a functional monomer. Polymer solution and initiator reagent were introduced to the micro-reactor system with varying flow rates, and principal component analysis (PCA) was employed to verify the optimum conditions. The continuously synthesized nanoparticles were collected as soon as 5 min of the process (288 times more efficient compared to the conventional method), and characterized by dynamic light scattering (DLS) and nanoparticle tracking analysis (NTA). Hence, we comprehensively benchmarked our system with various parameters such as channel length, flow rate, turnaround time, binding capability, and selectivity along with a detailed comparison with a conventional bulk production strategy. Considering the translation of imprinted nanoparticles into the market, the micro-reactor system minimizes the overall cost, as well as offers multiple-time use and higher yield compared to the conventional method.

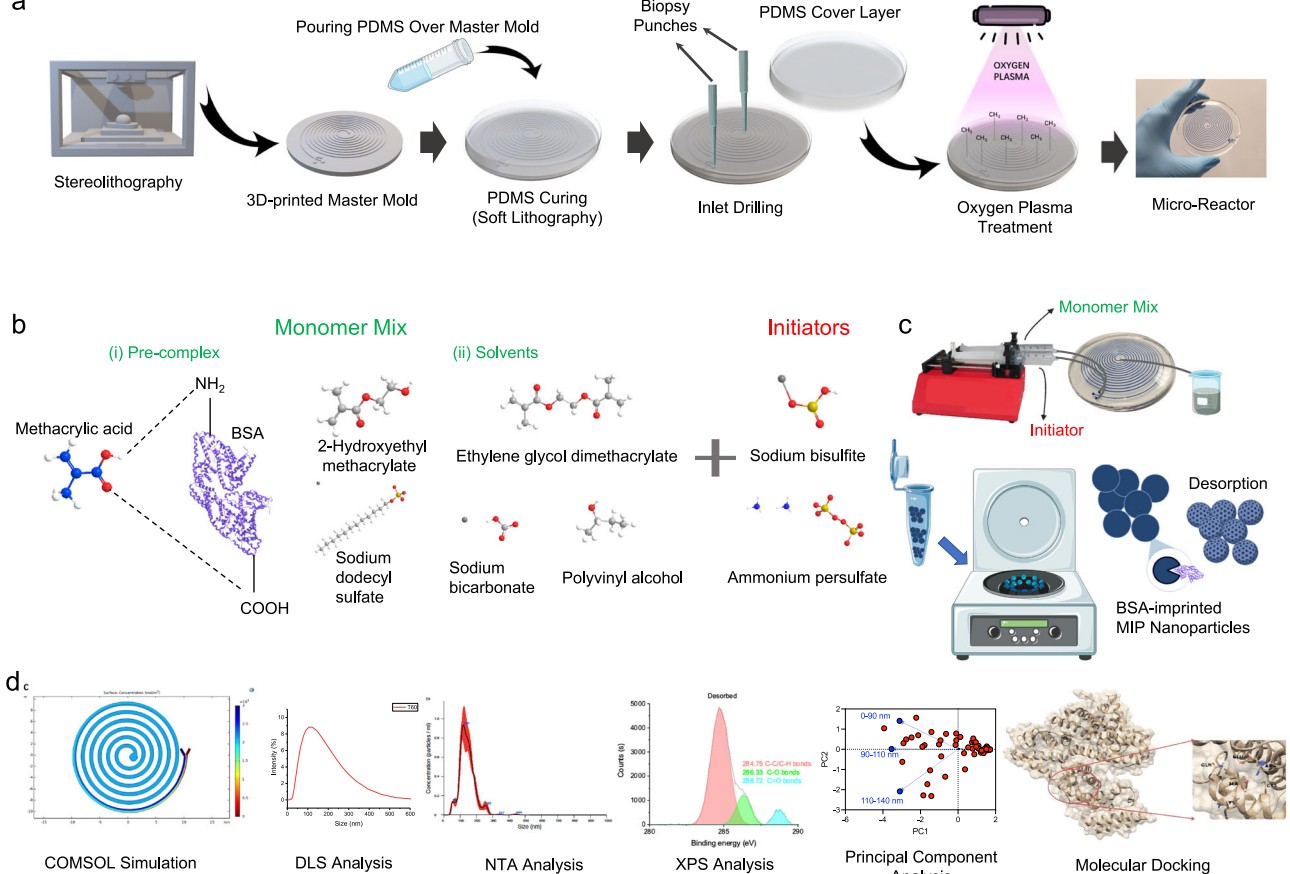

**Fig. 1 | Schematic representation of synthesis of BSA-imprinted nanoparticles on a micro-reactor. a** The fabrication of the system, **b** chemical composition of MIP nanoparticles, and **c** preparation and **d** characterization of BSA-imprinted nanoparticles are demonstrated, respectively. (The illustration in the figure **c** is created through Smart.Servier) (PDMS polydimethylsiloxane, BSA bovine serum albumin, MIP molecularly imprinted polymer, DLS dynamic light scattering, NTA nanoparticle tracking analysis, XPS X-ray photoelectron spectroscopy).

## Results

### COMSOL simulation results

The implementation of micro-reactors enables to produce nano-particles with adjustable physical and chemical properties, such as size, size distribution, and shape, resulting in uniform outcomes. In this regard, efficient mixing and polymerization through a chemical reaction were considered as the main criteria to acquire such prop-erties. Initially, considering the physical assets, the application of micro-reactors provides control over mixing parameters such as flow rate and ratio, resulting in efficient mixing. The way in which particles are mixed would also influence potentially both their structure and size distribution[94]. In addition to efficient mixing, the geometry of the micro-reactor would have impact on the polymerization process[95]. The design of microfluidic systems is highly influenced by the Reynolds number (Re) and the Péclet number (Pe), which enable precise control over the nanoparticle production[96,97]. On the other hand, while eval-uating the quality of nanoparticles produced on a micro-reactor, we cannot only consider efficient mixing and geometry characteristics since chemical reaction and duration time in the micro-reactor are the leading factors for the formation of nanoparticles. Accordingly, we considered simulation studies to fulfill the criteria of efficient mixing and geometry, and also, we evaluated the size and size distribution characteristics of nanoparticles as a quality criterion after performing experiments at the simulated conditions.

In this scenario, a computational method was utilized to system-atically solve for three physical processes involved, i.e., laminar flow, chemistry, and transport of diluted species. As a representative

purpose, we kept all the channel parameters same except the length 250 mm at each flow condition, thereby increasing the resolution in the simulation results. We also run the simulations for the micro-reactors with 1 m, 2 m, and 3 m of length (Supplementary Fig. 1) in order to mimic the same conditions in the experiments. Briefly, each process was solved step-by-step, with the solution of each step serving as the input for the subsequent step. Figure 2a shows the image of microchannels with two different fluids passing through. Figure 2b, c depicts the inlets for the monomer mixture and initiator, along with a corresponding color bar that illustrates the mixing efficiency of two fluids. The green color on the color bar represents a higher level of mixing efficiency. Next, in the chemistry module, chemical species for the monomer mixture and initiator were defined, along with their respective molar concentrations. The diffusion coefficients were calculated within the chemistry module through an automated process. This information was then utilized in the transport of diluted species to display the total concentration of these chemical species and their level of mixing effi-ciency. Figure 2d-f presents the relationship between various flow rates and the resulting mixing efficiency. As demonstrated, the closer the flow ratio was, the higher the mixing efficiency become. The flow rate of the monomer mixture was kept constant, while the flow rate of the initiator was gradually increased, resulting in an improved mixing effi-ciency observed in the comparison case of 143.75 vs 60 μL/min.

Per this observation, we also synthesized the MIPs according to the flow rates given in the simulation. According to our DLS data (Fig. 2g-i), the synthesized nanoparticles had a size range of 52–106 nm, 170–245 nm, and 85–175 nm for the initiator flow rates of

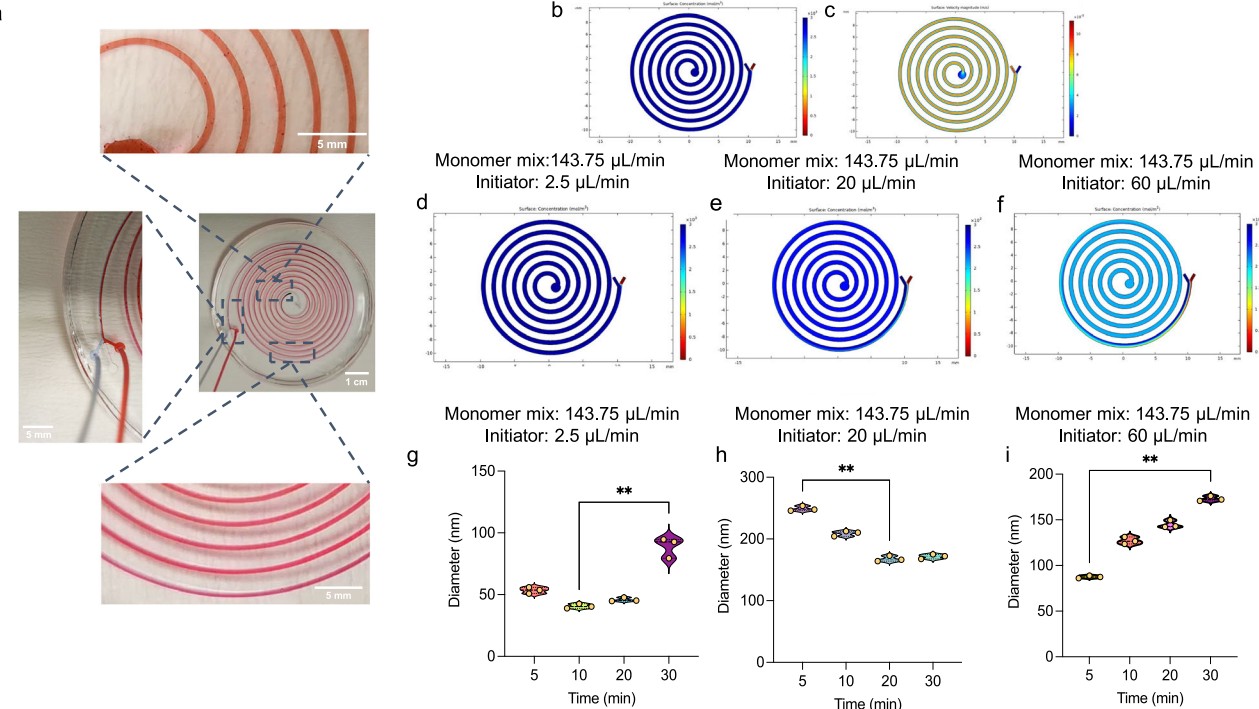

**Fig. 2 | Flow simulations of the micro-reactor and experimental data of the synthesized MIPs. a** The images of a micro-reactor with two different fluids (rhodamine 6G as red and commercial textile dye as light blue) passing through microchannels are represented. A schematic representation of microreactor demonstrates (**b**) surface concentration and **c** velocity magnitude profile, in which blue and red colors represent the monomer mix and the initiator, respectively. COMSOL simulation results state the surface concentration of MIPs synthesis using the combinations of flow rates that include (**d**) 143.75 µL/min and 2.5 µL/min; **e** 143.75 µL/min and 20 µL/min; **f** 143.75 µL/min and 60 µL/min. DLS data is presented for the combinations of flow rates, i.e., **g** 143.75 µL/min and 2.5 µL/min; **h** 143.75 µL/min and 20 µL/min; **i** 143.75 µL/min and 60 µL/min. The data is evaluated with one-way ANOVA (Freidman test) statistical analysis ($n = 3$), and the statistical difference is shown as $*p < 0.05$ and $**p < 0.01$.

2.5, 20, and 60 µL/min (while keeping the flow rate for the monomer mixture same), respectively. On the other hand, the average polydispersity index (PDI) values were 0.25, 0.295, and 0.182, respectively. Considering the PDI values, all the flow rates provided similar results; but the initiator flow rate of 60 µL/min resulted in more uniform structures. However, the size distribution of nanoparticles at different durations was mainly larger compared to the lower flow rates. Although the initiator flow rate of 60 µL/min led to more sequential increments in the nanoparticle size that stated more controllable production, this flow rate was not able to harvest lower diameters. This might have occurred since higher flow rates of initiator were not able to interact with monomer mixture for enough time in order to produce smaller particles. Another consideration would be that higher flow rates resulted in agglomeration and sticking nanoparticles, thereby increasing their sizes. Furthermore, the initiator flow rate of 20 µL/min enabled to produce larger sizes of nanoparticles (up to 245 nm), and it was not possible to produce at lower diameters (only down to 170 nm). The lowest flow rate of initiator (2.5 µL/min), on the other hand, resulted in yielding smaller diameters of nanoparticles (down to 52 nm) at the duration 5–20 min, and at the same time, enlarging the nanoparticle size was also possible at this flow rate while increasing the duration. Consequently, we proceeded our experiments using 1X flow rate conditions (the monomer mix flow rate of 143.75 µL/min and initiator flow rate of 2.5 µL/min) since smaller sizes of nanoparticles would have more surface area, thereby potentially interacting more numbers of target molecules on the course of sensing studies. For this flow rate, synthesis time was also further evaluated in the following sections.

## Computational results
Once COMSOL simulations were completed, we focused on understanding the interactions between target protein (BSA) and polymer matrix components (methacrylic acid (MA) and hydroxyethyl methacrylate (HEMA)) through molecular docking studies. We initially aimed to understand which position on BSA structure was reasonable for interacting with MA and MA-HEMA dimer, and accordingly, we performed a molecular docking study for these molecules.

Herein, we begun to demonstrate the complete structure of BSA (Fig. 3a), consisting of two main chains (A and B). Since these chains are identical, the interaction positions would be the same, and therefore, we only considered chain B of BSA (Fig. 3b). In this regard, we presented the coulombic surface coloring of MA, HEMA, and MA-HEMA dimer molecules. In particular, MA, HEMA, and MA-HEMA dimer have polar structures, and accordingly, their negative and positive charges are expanded on their own surface. This property hence leads to these three molecules to be surrounded mostly by polar residues, as well as the residues with positive and negative charges in the structure of BSA (Fig. 3d–f).

Once computing, we observed ten positions on BSA structure, where were favorable to interact with (i) MA and (ii) MA-HEMA dimer (Fig. 3c). Since in the experiments HEMA would interact to MA molecules to form a polymeric matrix, we first evaluated the most favorable positions on BSA structure for HEMA that were able to interact with MA-BSA complex while keeping the stability of the complex. Therefore, HEMA would be much closer to interact with MA easily, and would not be impeded with any steric hindrance. Accordingly, two positions (positions #1 and #4 in Fig. 3c) were considered since they were in the corner and close to the surface of the BSA structure, and thereby, HEMA would easily reach and interact to the complex (Fig. 3g and h). In the absence of HEMA, we also evaluated the favorable positions for MA-HEMA to interact with BSA structure, and position #5 in Fig. 3c would be a candidate for evaluating such interactions (Fig. 3i). As note, the remaining positions on BSA structure would have high

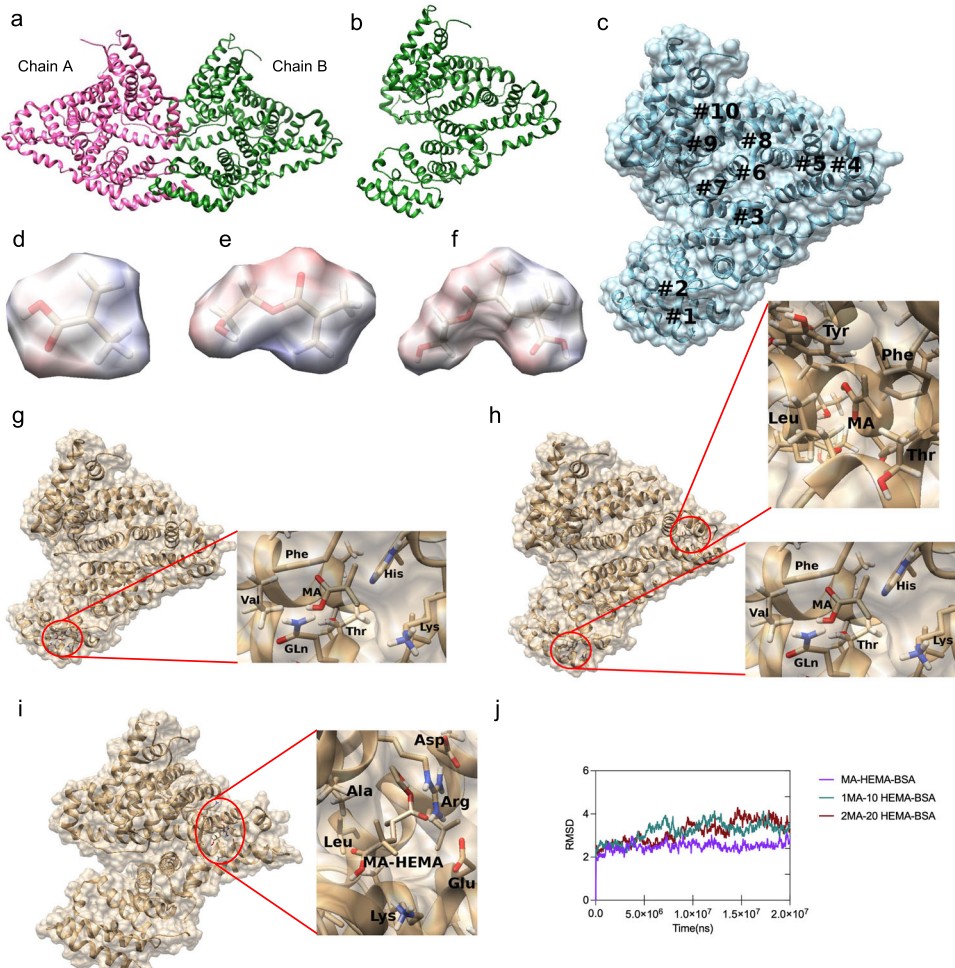

**Fig. 3 | Molecular simulations. a** Molecular structures of BSA and **b** chain B are demonstrated. **c** Molecular docking results present ten favorable positions for MA, interacting with BSA. **d–f** The coulombic surface coloring of MA, HEMA, and MA-HEMA are depicted. The red color indicates negative charges, whereas blue color shows positive charges and white color is neutral. **g** MA locates the surrounding residues in the position #1 on BSA. **h** MA interact the surrounding residues in both positions #1 and #4 on BSA. **i** MA-HEMA dimer interacts between the surrounding residues in the position #5 on BSA. **j** The RMSD values are calculated for three scenarios detailed in the manuscript (MA methacrylic acid, HEMA 2-hydroxyethyl methacrylate, BSA bovine serum albumin).

potential to interact with MA and MA-HEMA dimer since the value of their free energies resulted from the docking were similar to each one. Supplementary Table 1 presents the values of free energy [$\Delta G = (\Delta H - T\Delta S)$] for these ten positions. In this analysis, we only assessed the positions that HEMA would easily interact with the complex to form a polymeric matrix.

Then, we focused on the molecular dynamics (MD) simulations. In the experiments, MA and HEMA were used as approximately 1:10 molar ratio. The interactions of MA and HEMA with BSA were studied through the simulations of molecular dynamics, and we defined three major scenarios: (1) we evaluated the experimental conditions while considering 1 molecule of MA and 10 molecules of HEMA in the same condition. (2) We assessed any effects of molecule numbers over the interactions. Therefore, we computed 2 molecules of MA and 20 molecules of HEMA in the same condition while keeping the molar ratio same. (3) We investigated the interactions of MA-HEMA dimer with BSA, and also, compared this scenario with the other two scenarios.

Briefly, in the 1ˢᵗ scenario, 1 molecule of HEMA was able to interact with 1 molecule of MA, and however, the interaction energy was so low (−0.434 Kcal/mol). On the other hand, 4 molecules of HEMA were able to interact with the other potential binding positions on BSA structure, while the remaining HEMA (6 molecules) were moving around the

surface of protein freely. In the 2ⁿᵈ scenario, we observed that 14 molecules of HEMA were able to interact with the surface of protein, and the other 6 molecules were moving freely, pointing out that increasing the number of HEMA would have more chance to find favorable interactions on the protein structure. Likewise, we observed low interaction energy between MA and HEMA, stating the need for polymerization agent (ethylene glycol dimethacrylate (EGDMA)) as we used in the experiment. Considering this point, we examined the 3ʳᵈ scenario for understanding the MA-HEMA dimer interactions with BSA structure. On the other hand, HEMA has two functional groups, i.e., hydroxyl and carboxylic acid, and it would be expected that HEMA is able to interact with BSA structure easily; yet, its larger molecule size is hindered sterically. As the summary of the first two scenarios, HEMA prefers to locate in the same positions, where MA molecules also bind to BSA structure shown in Fig. 3c. These positions hold polar and charged residues such as His, Thr, Arg, and Lys in the inner side of BSA. We here presented the interaction energy [$E_{interaction} = (E_{van\ der\ Waals} + E_{electrostatic})$] of these molecules with BSA (Supplementary Table 2). Per the results in Supplementary Table 2, the interaction energy between MA-HEMA dimer and BSA was almost double than the other interactions (MA-BSA and HEMA-BSA). On the other hand, the affinity of MA-HEMA dimer presents more interaction energy because of the increased number of functional groups in dimer. This also indicates

that the first MA-HEMA dimer would have favorable interactions with BSA and can be an important base for the continuation of polymerization. While considering this outcome, we further investigated the effect of HEMA on the stability of BSA. Accordingly, we calculated the root mean square deviation (RMSD) of BSA for three scenarios (Fig. 3j). During the simulation time, we observed that the structure of BSA was completely stable while interacting with MA-HEMA dimer (in the absence of HEMA). Moreover, the RMSD value for the first two scenarios had more fluctuations and instability. Accordingly, these results confirmed that HEMA had the potential to interfere with the stability.

## Characterization of BSA-imprinted nanoparticles

Initially, we employed conventional synthesis of nanoparticles, and further physicochemical analysis of both MIP and non-imprinted polymer (NIP) nanoparticles was carried out with NTA and DLS analysis. In the conventional method, after 24 h of production, the average sizes of MIP and NIP nanoparticles were measured as $98.04 \pm 0.16$ nm and $161.2 \pm 0.19$ nm with low PDI values (0.16 and 0.19), respectively. At the same time of production, the particle concentrations for MIP and NIP nanoparticles were $1.87 \times 10^{13} \pm 1.38 \times 10^{12}$ particles/mL and $1.03 \times 10^{13} \pm 3.68 \times 10^{11}$ particles/mL, respectively. Consequently, MIP nanoparticles are anticipated to have smaller size and shape, in contrast to NIP nanoparticles. As we and the others observed in different synthesis methods[98,99], the size of MIP nanoparticles is typically smaller than those of NIP nanoparticles since the template molecule employed in the molecular imprinting process is able to possibly limit the size of the produced nanoparticles. In principle, functional monomers can form hydrogen-bonded dimers in the absence of a template, while synthesizing NIP nanoparticles, and the prepolymerization solution contains both functional monomer dimmers and free functional monomers. There are other possible molecular interactions between functional monomer and template in the MIPs that may potentially alter the formation of cross-linked polymer nuclei[100]. Although the preparation and polymerization processes were almost the same for synthesizing MIPs and NIPs, NIPs contained more functional monomer dimers in the pre-polymerization solution and the numbers of polymerization nuclei were generally less than in those of MIPs. This might have led to the size of the growing final cross-linked NIP nanoparticles nuclei to be significantly larger than that of the MIP nanoparticles. Moreover, the paucity of specific cavities on NIPs could have potentially caused them to have larger dimensions than MIPs. Since only the target molecule that fits inside the cavity was kept and the remaining one was washed away, the selective binding in the MIPs concluded to a decrease in the size of the nanoparticles[101].

On the other hand, the synthesis on our micro-reactor system resulted in slightly improved physicochemical properties of both MIP and NIP nanoparticles. The 1X flow rate (total flow rates (TFR): 146.23 μL/min) was determined as 143.75 μL/min for the monomer mixture and 2.5 μL/min for the initiators, adapted from the conventional method. The impact of total flow rate and microchannel length on the physicochemical properties of nanoparticles was analyzed to determine optimum synthesis conditions on the microreactor system.

To determine the effects of channel length on BSA-imprinted nanoparticle synthesis, three different channel lengths (1 m, 2 m, and 3 m) were examined at a constant TFR (146.23 μL/min). NTA and DLS analysis were performed to characterize the average size, PDI, and concentration of the nanoparticles collected at 30 min intervals (0-180 min) (Supplementary Figs. 2 and 5-10). Per statistical assessments, there was no significant difference between the average size of particles collected after each interval on micro-reactor synthesis using 1X flow rate − 1 m of channel length and 2X flow rate − 3 m of channel length (Fig. 4a and f) ($n = 3$, $p > 0.05$). On the other hand, we observed significant differences in particle sizes for certain intervals at the conditions of 1X flow rate − 2 m of channel length (Fig. 4b), 1X flow rate

− 3 m of channel length (Fig. 4c), 2X flow rate − 1 m of channel length (Fig. 4d), and 2X flow rate − 2 m of channel length (Fig. 4e). This might have occurred because polymerization happened at a slightly different rate through the micro-reactor. In addition, we did not observed any statistical differences between the groups at the conditions of 1X flow rate − 1 m of channel length and 2X flow rate − 3 m of channel length ($n = 3$, $p > 0.05$). Moreover, doubling TFR did not have an impact on the size of nanoparticles collected at the end of 180 min. In this study, we have tested only 1X and 2X flow rates and could not increase the TFR due to the limitations in the total volume of syringe. As the polymerization is strictly dependent on the interaction time between monomer and copolymer, increasing the flow rate would possibly reduce the interaction time and decrease the efficiency of polymerization[102].

According to the NTA results, the optimum condition that generated the smallest and the most monodisperse particles was determined under the conditions of 1X flow rate − 1 m of channel length. Herein, the average size of nanoparticles collected in 30 min intervals were $111.3 \pm 12.6$ nm, $104.6 \pm 7.3$ nm, $113.9 \pm 11.2$ nm, $106 \pm 6.9$ nm, $111.5 \pm 11.3$ nm and $106.6 \pm 16.4$ nm, respectively (Table 1), and all collected nanoparticles possessed more monodisperse characteristics than the other conditions. The concentration range of the collected nanoparticles were changed between $3.8 \times 10^{10}$ and $1.2 \times 10^{11}$ particles/mL (Supplementary Fig. 3). The sizes of nanoparticles obtained in the optimum selected condition were between 85 and 110 nm, along with a PDI value of -0.25. The average size of NIP nanoparticles was measured as $86.95 \pm 0.16$ nm, having a PDI of 0.28. As revealed in Fig. 4g and h, 30 min of synthesis with micro-reactor enabled to produce MIP and NIP nanoparticles with smaller in size compared to the conventional system. The size of MIPs and NIPs was found as $161 \pm 2.6$ nm, $247.9 \pm 2.1$ nm for conventional synthesis, while $88.94 \pm 8.3$ nm and $104.3 \pm 2$ nm were found for micro-reactor synthesis. The PDI values of MIPs and NIPs obtained in our micro-reactor (0.20–0.26) was lower than the PDI of nanoparticles synthesized by the conventional method (0.25–0.40).

After 30 min of production, the concentrations for MIPs and NIPs were calculated as $0.67 \times 10^{13} \pm 1.54 \times 10^{12}$ particles/mL and $1.04 \times 10^{13} \pm 3.57 \times 10^{11}$ particles/mL, respectively. These values for conventional synthesis were $1.36 \times 10^{13} \pm 2.83 \times 10^{11}$ and $1.03 \times 10^{13} \pm 7.55 \times 10^{11}$ particles/mL, respectively (Fig. 4i). The synthesis on the micro-reactor system produced lower number of particles than the conventional method since microfluidic strategy requires less material and energy use, and on the other hand, it allows in situ process control. In addition, when examining in terms of size, it was observed that smaller MIPs and NIPs could be synthesized compared to the conventional method.

Furthermore, we compared the workflow procedure of micro-reactor system with currently employed conventional method. For instance, as reported in the literature[2,3,103–105], the sequential centrifugation steps (4-5 times, and at the speeds of $14104 \times g – 55743 \times g$) are required in the conventional method to decrease the nanoparticle size. On the other hand, using the micro-reactor system, we centrifuged nanoparticles only once at low speeds ($2415 \times g$) to separate them from non-polymerizing materials and large particles. MIPs synthesized with the conventional method might have possess polydisperse properties due to reactions in a bulk environment. The purpose of sequentially repeated centrifuges is to separate particles of different sizes and to obtain the smallest particles in a monodisperse form. However, such a multi-step method is costly in terms of time and effort. With the micro-reactor system, we were able to synthesize particles within nm-scale with monodisperse assets without requiring the repeated centrifugation steps (only one single step of centrifugation) that decrease the process time and eventually increase cost-effectiveness of the process. Additionally, due to the affordable cost ($10 per chip), the micro-reactors can be placed in parallel to boost the productivity using a small area on a bench.

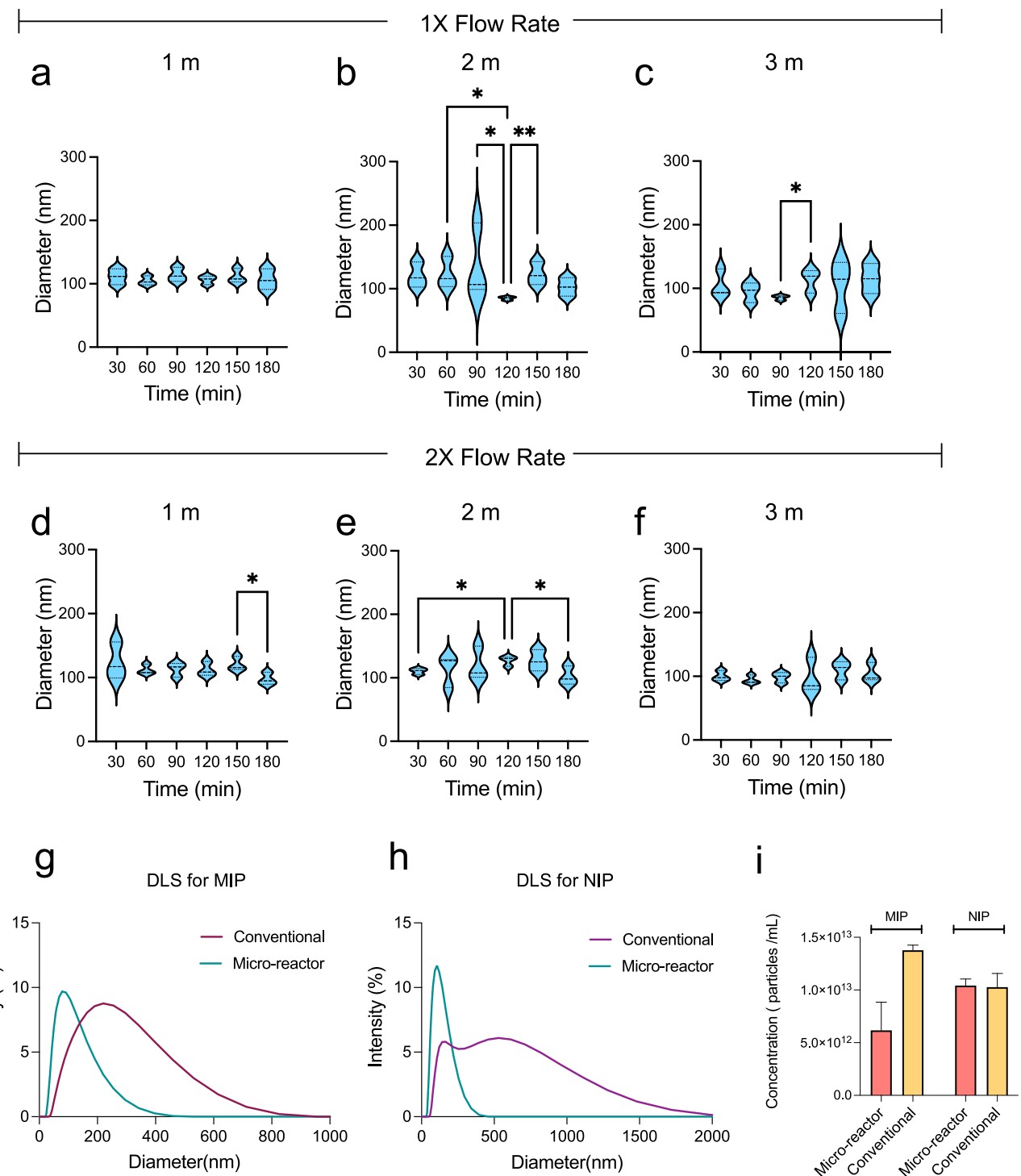

**Fig. 4 | Evaluating the parameters of micro-reactor and comparing the data with conventional method. a–f** The NTA analysis results are presented for BSA-imprinted nanoparticles with different channel lengths of micro-reactor. NTA results of MIPs synthesized with 1X flow rate are represented for **a** 1 m, **b** 2 m, and **c** 3 m, respectively. NTA results of MIPs synthesized with 2X flow rate are represented for **d** 1 m, **e** 2 m, and **f** 3 m, respectively. **g–i** DLS and concentration results of nanoparticles synthesized with conventional method and micro-reactor system (channel length: 1 m, flow rate: 1 X) for 30 min of the process are presented.

**g–h** Using the both methods, DLS results of MIPs and NIPs are demonstrated, respectively. **i** After 30 min of production, the concentrations of MIPs and NIPs synthesized with both micro-reactor and conventional method are compared. The error bars in the figure correspond to standard deviation of the mean. The data is evaluated with one-way ANOVA (Freidman test) statistical analysis ($n = 3$), and the statistical difference is shown as $*p < 0.05$ and $**p < 0.01$ (DLS dynamic light scattering, MIP molecularly imprinted polymer, NIP non-imprinted polymer).

We have also assessed shorter polymerization durations in both conventional method and micro-reactor production of MIPs. According to the results shown in Supplementary Fig. 4, conventional synthesis led to the formation of larger-size of MIPs (within the range of 150–200 nm), while MIPs collected from the micro-reactor system had smaller diameters (between 52 and 106 nm). In the micro-reactor, the interactions between the initiator and the monomer mixture might be higher due to the high surface-area-to-volume ratio, which resulted

**Table 1 | NTA and DLS results of BSA-imprinted nanoparticles collected at different time intervals**

| Time intervals (min) | 1m-1X | | 1m-2X | | 2m-1X | | 2m-2X | | 3m-1X | | 3m-2X | |
|---|---|---|---|---|---|---|---|---|---|---|---|---|
| | NTA (nm) | DLS (nm) | NTA (nm) | DLS (nm) | NTA (nm) | DLS (nm) | NTA (nm) | DLS (nm) | NTA (nm) | DLS (nm) | NTA (nm) | DLS (nm) |
| 30 | 111.7±12.6 | 88.94±8.3 | 123.3±28.7 | 87.6±5.6 | 120.6±19.8 | 101,29±57.4 | 110.3±4.8 | 116.4±22 | 105.6±21.6 | 138.5±25 | 100.2±7.9 | 105.1±8.4 |
| 60 | 104.6±7.3 | 92.36±9.8 | 112.1±7.8 | 110.8±26.3 | 123.3±24.5 | 102±54.7 | 113.4±25.06 | 132,3±14.5 | 94.3±15.5 | 141.9±27.9 | 94.4±6.6 | 107.1±1.7 |
| 90 | 113.9±11.2 | 99.9±8.9 | 111.3±11.2 | 95.06±6.4 | 136.4±58.4 | 106.9±65.4 | 119.5±26.5 | 140.3±53.75 | 85.5±3.9 | 126.9±26.2 | 98.4±8 | 115.4±6.4 |
| 120 | 106±6.9 | 105.5±10.6 | 112.3±11.3 | 105.5±1.3 | 85.03±2.82 | 114.92±64.3 | 126.9±7.8 | 155.2±10.7 | 113.2±18.3 | 163.8±12.9 | 98.06±27.62 | 113.2±7.2 |
| 150 | 111.5±11.3 | 103.75±8.5 | 120.5±11.0 | 111.9±6.9 | 123.23±18.09 | 107.46±53.2 | 102.3±14.7 | 118±18.1 | 105.2±40.8 | 172.5±13 | 110.3±14.5 | 131.5±25.1 |
| 180 | 106.6±16.4 | 88.18±8.2 | 97.5±9.6 | 99.9±23.3 | 102.7±14.5 | 119±31.5 | 126.7±16.8 | 134.1±20 | 115.2±23.6 | 192.8±25.2 | 104.6±15 | 120.9±24.7 |

Abbreviations: "m" refers to channel length of micro-reactor (1 meter and 2 meters); "x" refers to flow rate (Total flow rate: 1X = 146.23 µL/min and 2X = 292.46 µL/min).

in the synthesis of smaller MIPs. The collective dry-weight of MIPs were also calculated (Supplementary Table 3), and at all the durations, the microreactor system provided 1.4–1.5 times more MIPs compared to the conventional method, and with our platform, we were able to produce 263.8 mg of MIPs within 180 min.

## Principal component analysis
After collecting all the data for nanoparticle synthesis using the micro-reactor system, we defined the first principal component as the greatest amount of variance within the data. The second principal component accounted for the second greatest variance that was orthogonal to the first principal component. Per the PCA results, the projection (loadings) of the target interval (i.e., 90–110 nm) was closer to the first principal component for F1 (Flow rate of 1X). In Fig. 5a, the length M = 1 stated the target range closest to the first principal component. Based on these results, both M1 and F1 were the optimal conditions in order to obtain the highest yield of nanoparticles within the target range. F1 and M1 also complied with the experimental results (Fig. 5a-c).

The difference in the distribution of PC scores among the different methods is noteworthy. In F2, the PC scores aligned more firmly with the second principal component (Fig. 5b). In the case of F1, they more loosely aligned with the first principal component. Similar behavior was also observed in M1, M2 and M3, where the PC scores of M2 aligned more firmly with the second principal component, whereas in M1 and M3, they agreed more (albeit loosely) with the first principal component (Fig. 5c-e).

## Characterization studies
Once finalized to synthesize nanoparticles on the micro-reactor system, we designed a binding study on a plasmonic metamaterial sensor from an application perspective (Fig. 6a). Before the binding tests, topographical analyses of MIPs and bare sensor were initially assessed with atomic force microscopy (AFM) studies while they were attached on the metamaterial sensor including nanoperiodic structures (Fig. 6b-e). The surface depth and surface roughness were observed as 60.1 nm and 14.16 nm for the MIPs immobilized sensor, while it was calculated as 128.5 nm and 52.84 nm for the bare surface, respectively. These results also showed successful attachment of MIPs to the sensor, and also, this attachment led to the roughness value as ~38 nm, stating the potential coverage in the grooves of the nanoperiodic arrays on the sensor.

Later, we performed an X-ray photoelectron spectroscopy (XPS) analysis to characterize the elemental composition and chemical state in the BSA-imprinted and desorbed nanoparticles. The chemical composition of the nanoparticles was examined with C1*s*, N1*s*, and O1*s*. The XPS spectra of desorbed and non-desorbed nanoparticles was depicted in Fig. 6f–k. The elemental composition of both nanoparticles includes three types of carbon atoms corresponding to C-C/C-H at 284 eV, C-O 286 eV and C = O at 288 eV (Fig. 6f, g) and two types of oxygen atoms corresponding to C-O at 532 eV and C = O at 532 eV (Fig. 6j, k). When BSA protein was desorbed from MIPs, the intensity of peaks corresponding to C = O bonding decreased due to the reduction of carbonyl groups on BSA.

## Adsorption-desorption and binding studies
Before benchmarking on a metamaterial sensor, imprinted nanoparticles were evaluated for adsorption-desorption cycles. According to the absorbance measurement at 280 nm, BSA protein was successfully desorbed from the nanoparticles and adsorbed again. As shown in Supplementary Fig. 11a, b, their absorbance values decreased from $0.332 \pm 0.005$ to $0.028 \pm 0.004$ arb. units, after the desorption, and increased again up to $0.249 \pm 0.002$ arb. units.

Afterward, the MIPs were immobilized on a metamaterial sensor, and various concentrations of BSA solution from 10-50 µM were applied to the sensor. Higher concentrations of BSA solution led to increase the plasmonic signals (wavelength shifts) (Fig. 7a-e). After an equilibrium step with sodium acetate buffer at pH 4, various concentrations of BSA solution were introduced to the system. Then, a washing step was taken place in order to remove unbound molecules. BSA protein efficaciously bound to the MIP nanoparticles on the sensor at all concentrations. The subsequent desorption step showed a blue-shift due to the difference in the refractive index of the NaCl solution. After desorption with 0.5 M of NaCl, the system was re-washed with the buffer at pH 4. According to the entire analyses, the removal step of the protein from the MIP nanoparticles has been accomplished successfully, and the system has returned to the equilibrium. The performance of BSA-imprinted nanoparticles represented 81% of precision for BSA solutions within concentrations of 10-50 µM, and the sensor provided a linear correlation within the concentrations of BSA solution applied (Fig. 7f).

## Selectivity, imprinting factor, and repeatability analysis
Briefly, selectivity analysis compares the binding effects of structurally analogue molecule (human serum albumin: HSA) with the target molecule (BSA). Hence, the selectivity performance of MIP nanoparticles synthesized on the micro-reactor system was evaluated, while testing their binding with HSA protein (20 µM) due to molecular similarity in the chemical compositions. As shown in Fig. 7g, a wavelength shift of $0.9 \pm 0.1$ nm occurred when BSA protein (20 µM) was introduced into the system, while this value was $0.2 \pm 0.1$ nm for HSA (20 µM). The selectivity coefficient (*k*) was calculated according to Eq. 1 as follows:

$$\mathbf{k} = Shift_{BSA}/Shift_{HSA} \qquad (1)$$

Considering the earlier reports[106], we calculated the k value as 4.5. If k value is greater than 1, it demonstrates high affinity of MIPs to the target molecule, i.e., BSA protein[3]. In addition, imprinting factor (relative selectivity coefficient (*k'*)) defines the effect of molecular grooves for the binding of target molecules, thereby comparing the binding impact of MIPs and NIPs. The *k'* value was calculated using the Eq. 2.

$$\mathbf{k}' = \mathbf{k}_{MIP}/\mathbf{k}_{NIP} \qquad (2)$$

While applying 30 µM of BSA solution to both MIPs and NIPs, we observed that the response for the NIPs was $0.9 \pm 0.3$ nm, and this value was found to be $2.1 \pm 0.5$ nm for MIPs. Hence, the imprinted

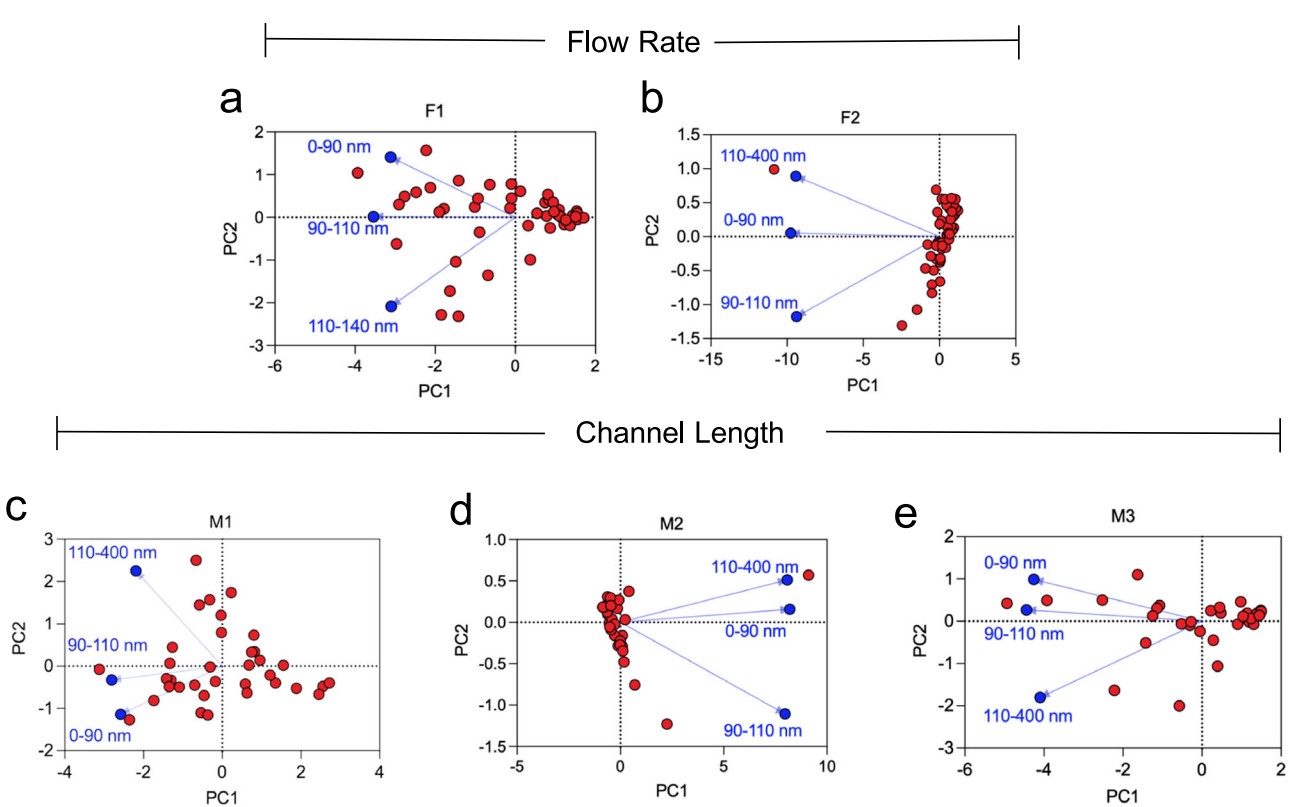

**Fig. 5 | Principal component analysis of micro-reactor synthesis of imprinted nanoparticles. a, b** The plots show the principal component analyses results obtained at (**a**) the flow rates of 1X (F1) and **b** 2X (F2), while (**c**–**e**) the other plots exhibit the results of different channel lengths that include (**c**) 1 m (M1), **d** 2 m (M2), and **e** 3 m (M3).

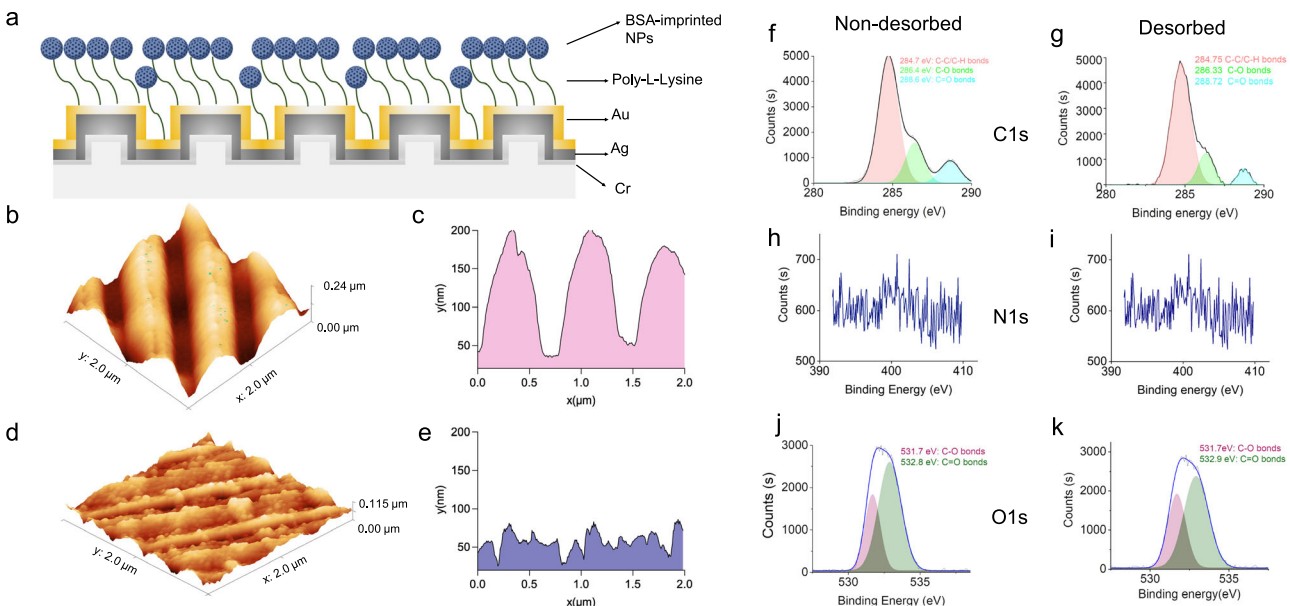

**Fig. 6 | Characterization studies. a** The schematic depicts the schematic illustration of MIPs immobilized on a plasmonic metamaterial sensor. **b**–**e** AFM images of plasmonic metamaterial sensor surfaces are presented. **b, c** The plots represent the bare surface, whereas **d, e** the other ones depict the MIPs-immobilized sensor surface. **f**-**k** The XPS analysis shows non-desorbed (**f, h,** and **j**) and desorbed (**g, i,** and **k**) nanoparticles with corresponding chemical groups, and their binding energies are stated on the plots for **f, g** C1s, **h, i** N1s, and **j, k** O1s (BSA-imprinted NPs bovine serum albumin-imprinted nanoparticles, Au gold, Ag silver, Cr chromium).

nanoparticles were able to detect BSA proteins with 2.3 times greater than the NIP nanoparticles (Fig. 7h).

Moreover, the repeatability performance of the imprinted nanoparticles was examined using the same concentration of BSA solution (30 μM) in three adsorption-desorption cycles on a sensor. The wavelength shift was decreased from 2.4 ± 0.3 nm to 1.75 ± 0.3 nm, and there was only 0.65 nm loss of performance (Fig. 7i). We here demonstrated that the MIPs synthesized on our micro-reactor and

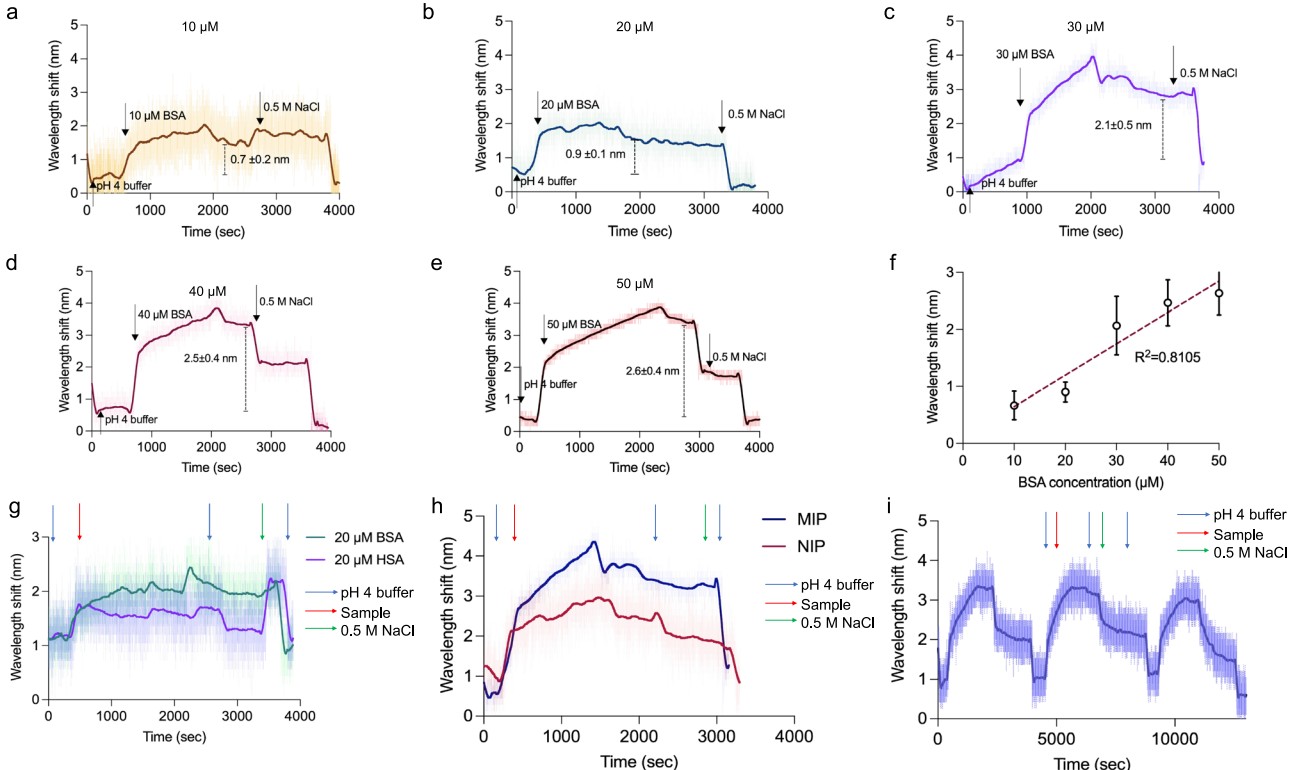

**Fig. 7 | Binding studies. a-f** The plots display the binding results of BSA protein solutions at different concentrations: **a** 10 µM, **b** 20 µM, **c** 30 µM, **d** 40 µM, and **e** 50 µM. **f** The calibration curve is plotted for evaluating the plasmonic signals upon the increments of protein concentrations. **g** As a selectivity analysis, the binding of BSA and HSA proteins is evaluated on the sensor modified with MIP nanoparticles. Wavelength shifts are collected in real-time data for BSA and HSA solutions at the same concentrations (20 µM for both protein solutions), and this plot states the selective binding of BSA protein to the MIPs. **h** Imprinting factor is evaluated through the real-time responses collected for the binding of BSA (30 µM) to MIPs and NIPs. **i** Three times of adsorption and desorption cycles are evaluated by introducing 30 µM of BSA and 0.5 M of NaCl as a desorption buffer. The error bars in the figure correspond to standard deviation of the mean. Blue arrow represents the buffer at pH 4.0, while red arrow represents protein solutions and green arrow represents NaCl solution (BSA bovine serum albumin, HSA human serum albumin, NaCl sodium chloride).

their mutual integration with the sensor worked well in detecting target proteins for several times with negligible performance lost.

## Discussion

In this study, we developed a micro-reactor for the continuous synthesis of imprinted nanoparticles. Initially, we performed COMSOL simulation, and further experimentally analyzed a number of flow rates for the initiator (20 and 60 µL/min) to synthesize MIPs at these conditions. The distribution of nanoparticle sizes was found to be consistently larger at different time intervals as compared to the smaller sizes observed at lower flow rates. Although the initiator flow rate of 60 µL/min formed nanoparticles with sequential size increments, it was unable to produce smaller-sized nanoparticles. Considering the higher surface area, we successfully produced small nanoparticles down to 52-106 nm at the initiator flow rate of 2.5 µL/min. Afterward, the micro-reactor system was comprehensively benchmarked by optimizing essential parameters such as channel length, flow rate, turnaround time, binding capability, and selectivity. The flow rate of 146.23 µL/min and channel length of 1 m were defined as the optimum conditions after the assessments with nonparametric Freidman test and PCA. Under these conditions, the nanoparticle size was around 100 nm starting from 30 min of synthesis and provided a concentration of $0.67 \times 10^{13} \pm 1.54 \times 10^{12}$ particles/mL for MIPs and $1.04 \times 10^{13} \pm 3.57 \times 10^{11}$ particles/mL for NIPs. The micro-reactor system enables in situ control of MIPs synthesis by tuning the channel length, flow rate, and assay time, thereby allowing to produce MIPs with distinct diameters. On the other hand, the conventional method holds considerable stumbling blocks in yield since any alterations in

experimental condition would potentially hinder the polymerization and growing of MIPs, and lead to produce polydisperse particle. Hence, in situ control of synthesis is a critical need for both reproducibility and high-yield; yet, the conventional method cannot allow such a control/feedback loop.

Afterward, a computational analysis was also carried out to discover interactions between monomer (MA)/co-monomer (HEMA)/MA-HEMA dimer and target protein (BSA). Low interaction energy between MA and HEMA indicated the need for a polymerization agent (such as EGDMA) to enable sufficient interactions between these molecules. HEMA preferred to interact with the inner positions of BSA because of potential accumulation of polar and charged residues such as His, Thr, Arg, and Lys in the inner site. In addition, the presence of HEMA moving freely around BSA that could not find favorable positions to interact, impede the stability of BSA. Considering three scenarios, MA-HEMA dimer had the most favorable interaction energy due to the number of functional groups in the structure.

In addition, the presented system enables the collection of imprinted nanoparticles within 5-30 min; and minimizes the overall cost down to $10 per chip, which can also be used multiple times. Besides that, in every 30 min of time intervals, the micro-reactor system provided 1.4-1.5 times more MIPs compared to the conventional method according to the calculated dry weights. On the other hand, the counterpart strategy—conventional method, required 24 h to produce the intended size of imprinted nanoparticles ( ~100 nm). In addition, the use of the micro-reactor is very facile for proper operation, including: (i) filling two syringes with samples (one for polymer mixture and the other for initiator) and (ii) running the samples under

a controlled flow rate on a pump. For this scheme of operation, a standard syringe pump is the only requirement that would be further miniaturized with existing open-source systems with a total cost under $100 (i.e., Bartels mp6 micropumps). Considering easy adaptation of this strategy to produce MIPs in an efficient manner, pressure/vacuum controllers and generators, as supplied, for instance, by ElveFlow®, would be a one-time purchase, and the micro-reactor would provide more controllable process in producing more complex synthesis in situ. Furthermore, the total volume for the synthesis relies on the number and size of imprinted nanoparticles planned. Considering the requirements of reagent volumes, for the micro-reactor system, we only run 20 μL of MA solution (monomer), 0.25 mL of HEMA solution (co-monomer), 0.5 mL of EGDMA solution, and 80 μL of BSA solution (in pH 4 buffer) (as a model target), whilst the conventional method required 40 μL of MA solution (monomer), 0.5 mL of HEMA solution (co-monomer), 1 mL of EGDMA solution, and 160 μL of BSA solution, therefore minimizing the volume of reagents down to 50%.

MIPs can be synthesized through various methods, such as bulk, precipitation, emulsion, suspension, core–shell polymerization, and electrochemical methods. Briefly, in bulk imprinting technique, target analyte is incorporated directly into the polymer matrix[107]. In this conventional imprinting method, monomers and crosslinkers are polymerized in the presence of template molecules by thermal/photo initiation, and then, a bulk polymer is formed[108]. However, this strategy utilizes large volume of reagents and takes lengthy processes. Another technique used for the fabrication of MIPs is emulsion polymerization, which enables the synthesis of monodispersed MIP nanoparticles containing surface-exposed binding sites. Polymerization typically takes place in the existence of a surfactant in oil-in-water emulsions[109], and the monomers is commonly dissolved in an aqueous solution without the use of surfactants or emulsifiers[110]. In this method, the binding sites on the surface of imprinted microspheres or nanoparticles are evenly distributed, and the reuse rate of the MIPs obtained is high[111]. Yet, the use of water and surfactants during polymerization between template and functional monomer can cause precipitations potentially[112]. Another frequently used process is suspension polymerization. Uniform and similarly sized microspheres are obtained from small colloidal drops of the polymerization mixture suspended in a liquid phase with this method. The purpose of suspension polymerization is to obtain a homogeneous distribution of round-shaped MIPs with sufficient binding properties[113]. However, controlling particle size distribution is a challenge in suspension polymerization since the liquid droplet size distribution depends on the type and concentration of surfactant, the quality of agitation, and physical properties, such as density and interfacial tension[114]. Besides, using the core-shell structure to increase MIP performance is an appealing option. The surface-imprinted shell assures that the bonding sites on the surface are evenly distributed. This analysis results in faster and more efficient template molecule removal and reattachment with enhanced binding kinetics. Such an approach provides MIPs with uniform and predetermined particle sizes. Moreover, MIP coated core-shell nanoparticles have become multifunctional with magnetic, optical or semiconducting properties of the inorganic core[115]. However, there are some considerations in this process: (i) most of the interactions of macromolecules (i.e., proteins) with other molecules occurs in water and not in the organic system. (ii) Core-shell synthesis in aqueous media needs to be well investigated while applying this method for imprinting of macromolecules. (iii) Ideally more biocompatible and hydrophilic functional monomers that can reduce the effect of hydrogen bonding needs to be preferred[116]. The latter, electrochemical method, is particularly employed for conducting polymers, and it relies on multiple parameters including: (i) applied voltage, (ii) potential sweep rate, (iii) the control of charge, (iv) the duration and periods of applied potential pulses, (v) any external treatment such as ultrasound, and (vi) the variations in ion and material concentrations[80].

Therefore, thickness, ion permeability, density, and porosity are precisely controlled by tunning such parameters. On the other hand, overoxidization of polymers is crucial for forming MIPs, and such polymeric structures can be applied for (1) generating oxidized radicals in order to rise sensitivity and/or selectivity towards target molecules, and (2) facilitating template removal and/or regeneration of MIP-based structures[80]. Among aforementioned conducting polymers, polypyrrole can be easily synthesized through chemical and electrochemical methods, and it benfits more from overoxidization fashion in forming MIPs. Yet, there would be more attempts to improve and broaden the applications of polypyrrole-based MIPs in the field. On the other hand, polypyrrole exhibits notable compatibility with biological entities and does not irritate immune system of mammalians, thereby holding great potential to be applicable in developing implantable biosensors in the near future.

Among biosensing modalities, MIPs are extensively utilized in electrochemical and optical detection systems. As a proof of concept, in this study, we evaluated the binding performance of MIPs synthesized on the micro-reactor using optical sensors (plasmonic metamaterial sensor). Considering their use in biosensing, imprinted nanoparticles synthesized on the chip have achieved to detect various concentrations of BSA solution (10-50 μM) with 81% of sensitivity, and resulted in 4.5 times more selectivity while testing with a molecularly-similar protein (HSA). Comparing MIPs with NIPs in terms of biosensing, the imprinted nanoparticles were able to detect BSA proteins with 2.3 times greater than the NIP nanoparticles, pointing out that the need of specialized grooves for detecting BSA proteins. On the course of selectivity, multiple-time use had a negligible loss of performance (0.65 nm of plasmonic signal) after three consecutive uses.

The remaining challenges associated with the micro-reactor synthesis of imprinted nanoparticles and efficiency of the entire system include limitations in scale-up due to the miniaturized nature of microfluidic devices. Parallelization, which is placing many replicates micro-reactors that operate in parallel, can be proposed as a successful strategy to increase the throughput of the MIP synthesis. Polymerization is the process in which small molecules combine to generate long-chain polymers, and it is highly dependent on the concentration of the initiator and the monomer mixture. These parameters should be defined accordingly in order to avoid blockage problem in the microchannels, as they may directly impact the polymerization rate in the microreactor. In our study, we have considered the parameters used in the bulk process and adapted them to our micro-reactor system for a side-by-side comparison. While applying polymer solutions, there would be two major possibilities to hinder the efficiency of micro-reactor. (i) Clogging/blockage of channels in micro-reactor would occur since there would be possible smears of polymers on PDMS surface of channels. (ii) Though we did not observe in our process, polymer solutions would react to the microchannel surface. The critical point is that both solutions should simultaneously be introduced to the micro-reactor. This way, more uniform polymerization could be provided potentially. However, due to different flow rates of both solutions (2.5 μL/min and 143.75 μL/min), it was highly challenging to introduce them into the system at the same time. When the initiator was introduced into the micro-reactor before the monomer mixture, we observed some accumulations of initiator in the initial part of the main channel, which resulted in intense polymerization upon meeting the monomer mixture. The critical point is that both solutions should be introduced simultaneously to the micro-reactor. This way, more uniform polymerization could be provided potentially.

Durability and long-term stability are highly dependent on polymers utilized in MIP formation. HEMA-originated polymeric matrix used in this study has been tested for long-term durability and stability by us and the others in earlier studies[117,118], and therefore, we did not perform such evaluations here. As an example, MIPs were challenged for

36 months of incubation at room temperature after they were coated on the sensors[118]. This long-term storage did not cause a statistically significant change in their performance, and the MIP-coated sensor response only changed around 15%. Likewise, the researchers prepared synthetic cannabinoids imprinted polymers, and conducted stability analysis for 12 months, and there was no significant decrease in the sensor's performance[117]. Moreover, lyophilization, often known as freeze-drying, can be used to increase the durability of MIP nanoparticles[119]. On the other hand, long-term MIP synthesis was performed in the micro-reactor for 180 minutes, and we did not observe any performance issues for producing trillions of MIP nanoparticles on a single device. Evaluating the performance of micro-reactor for longer periods than 180 minutes would be beneficial for synthesizing more particles on a single batch. Last but not least, the proposed micro-reactor system stands out as a suitable alternative to the conventional counterparts for MIP synthesis for being cost-effective streamlined approach.

## Methods

### Fabrication of micro-reactor system
Negative templates to fabricate micro-reactor with varying channel lengths (1, 2, and 3 meters) were designed using Autodesk Inventor Professional 2021. The channel height and width of micro-reactor were designed to be 500 and 600 μm, respectively. The template was then printed in a desktop 3D printer "Form 3" (Formlabs GmbH, Germany) using stereolithography (SLA) technique. Clear resin (Formlabs GmbH, Germany) was employed to 3D print the template with a layer thickness of 25 μm. Non-cured resin residue after SLA process was first immersed in isopropyl alcohol (99.9%) for 15 min as a solvent treatment followed by heat and UV-light treatments at 60 ℃ for 15 min in Creality UW-01 Washing/Curing Machine (Creality, China). After post curing process, the negative template was covered with high temperature-resistant Kapton polyimide film (Dupont, USA). PDMS with curing agent at ratio of 10:1 was prepared and poured over the 3D-printed negative template with channels and cover template without channels for soft lithography processing. Afterward, the templates were kept in the oven at 80 ℃ for 2 h to cure PDMS. PDMS layers were peeled off from the templates, and three inlets and one outlet were opened using biopsy punch with a diameter of 1 mm. Finally, two PDMS layers were irreversibly sealed using $O_2$ plasma-induced surface activation (50 sccm $O_2$, RF power of 100 W, chamber temperature of 25 ℃ and cycle time of 2 min). Schematic illustration of micro-reactor fabrication is shown in Fig. 1a.

### COMSOL Simulation
In this study, COMSOL Multiphysics 5.6 was utilized to analyze the mixing efficiency with different flow rates for monomer mixture and initiator through the integration and solution of three physics, i.e., laminar flow, transport of diluted species, and chemistry. A controlled flow rate was initially established to attain stabilized conditions for the fluid flow. Upon establishing the stabilized conditions, the mixing location within the channel was analyzed, and we observed that the downstream length at which mixing occurred remained constant over time. As a result, it was established that the mixing location was independent of time, and the stationary mode was selected accordingly. A computational method was then utilized to systematically solve for three physical processes involved: laminar flow, chemistry, and transport of diluted species. Herein, we have initially run our analysis using a 2D model, which had same thickness and width values of the micro-reactor, and differed in the length value only (250 mm) in order to depict more resolution in the figures. This model precisely represented the micro-reactor designed in our study. In addition, we have also evaluated the real geometries used in our experiments including exact length (Supplementary Fig. 1).

### Computational analyses
Docking and molecular dynamic simulation were performed to understand the interactions between the monomers and BSA (Supplementary Materials and Methods 1.2).

### Micro-reactor synthesis of BSA-imprinted nanoparticles
For the micro-reactor synthesis of BSA-imprinted nanoparticles, same aqueous phases used in the conventional synthesis were prepared. After homogenization process, monomer mixture (which includes EGDMA, HEMA, and pre-complex solutions) was gently shaken and transferred to a sterile 50 mL syringe. Initiators including ammonium persulfate and sodium bisulfite were prepared and transferred to a sterile 5 mL syringe. Monomer mixture and initiators were introduced to the micro-reactor system with the determined flow rates for each solution. Polymerization was carried out at 40 ℃ in order to enable same conditions in the conventional method. In order to determine the effect of different flow rates on BSA-imprinted nanoparticle synthesis, two different TFRs (1X = 146.23 μL/min and 2X = 292.46 μL/min) were examined. In addition, the effect of the channel length of the micro-reactor on particle synthesis was also investigated. In this context, devices with different channel lengths, (1 m, 2 m, and 3 m) have been prepared. Samples were collected every 30 min and stored in the fridge (4℃) for further characterization. NIPs were also prepared using the same protocol without using the solution of BSA. The measurement of each parameter was conducted in triplicates. Additionally, in order to test shorter polymerization time, we also collected MIPs at the shorter intervals (5, 10, and 20 min) for both conventional method and micro-reactor system. The collective dry-weight of MIPs per mL collected between 30 and 180 min intervals were also calculated after the MIPs were lyophilized. The desorption of BSA protein solution from MIP nanoparticles was carried out using 0.5 M NaCl for an hour until no absorbance was observed at 280 nm.

### Conventional synthesis of BSA-imprinted nanoparticles
In order to benchmark the performance of our strategy, we implemented a conventional method to produce imprinted nanoparticles in a batch process, where a homogenizer and a large reaction flask (250 mL) were connected to each other, and the synthesis took for 24 h. Here, we employed BSA as a model protein to imprint into the polymer matrix through both conventional and micro-reactor methods (Supplementary Materials and Methods 1.3).

### Characterization of BSA-imprinted nanoparticles
Physiochemical and chemical characterization of BSA-imprinted and NIPs was performed using DLS, NTA, and XPS (Supplementary Materials and Methods 1.4).

### Principal component analysis (PCA)
PCA was applied to the data obtained from NTA to assess the findings statistically and reduce the dimensionality of the data for a better interpretation (Supplementary Materials and Methods 1.5).

### Binding analysis
Preparation and characterization of plasmonic metamaterial sensor, adsorption-desorption, kinetic, selectivity and repeatability studies were explained in detail in Supplementary Materials and Methods 1.6-1.10.

### Statistical analysis
All collected data were statistically analyzed (Supplementary Materials and Methods 1.11).

## Data availability
Data that support the findings of this study are available from the corresponding author upon request.

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

## Acknowledgements

Drs. Fatih Inci, Ismail Eş, and Yeşeren Saylan gratefully acknowledge the support of TÜBİTAK 2232–International Fellowship for Outstanding Researchers (project no: 118C254). This publication has been produced benefiting from the 2232 International Fellowship for Outstanding Researchers Program of TÜBİTAK (project no: 118C254). However, the entire responsibility for the article belongs to the owner of the article. The financial support received from TÜBİTAK does not mean that the content of the publication is approved in a scientific sense by TÜBİTAK. Dr. Fatih Inci also acknowledges the financial support of TÜBİTAK 3501–Career Development Program (CAREER) (project no: 120Z335), and GEBIP Award from Turkish Academy of Sciences (TÜBA). Dr. Özgecan Erdem gratefully acknowledges the support of the SANCAR Fellowship from Hacettepe–Bilkent University–UNAM and TÜBİTAK 2218-National Postdoctoral Research Fellowship Program (project no: 121C431). Drs. Yeşeren Saylan and Maryam Atabay are thankful for the support of TÜBİTAK 2247-D National Early-Stage Researchers Program (project no: 121C226). Murat Alp Gungen acknowledge the scholarship from Bilkent University - UNAM (University Graduate Fellowship). This work was also supported by BAGEP Award of the Science Academy. The numerical calculations reported in this paper were fully performed at TÜBİTAK ULAKBİM, High Performance and Grid Computing Center (TRUBA resources). All authors gratefully acknowledge this resource support. All authors greatly acknowledge the infrastructure, facilities, and continuing support from Bilkent University - UNAM. Lastly, this work is dedicated to the memory of Prof. Dr. Fatma Neşe Kök, who passed away on May 28, 2022, and we were able to only say "angels deserve to die" after her passing.

## Author contributions

Ö.E. synthesized and performed the characterization of MIP nanoparticles, and carried out the studies related to plasmonic sensors. I.E., fabricated the micro-reactor and contributed to synthesis and characterization of nanoparticles. Y.S. synthesized the MIP nanoparticles. M.A. performed the computational analysis. M.A.G. performed the PCA analysis. K.Ö. performed the COMSOL simulations. A.D. contributed to the concept of the manuscript, and provided the resources. F.I. generated the conception of the work/idea and supervised the research. Ö.E., I.E, Y.S., M.A., M.A.G., K.Ö., and F.I. contributed to write and edit the manuscript. All authors discussed the results and commented on the manuscript.

## Competing interests

The authors declare no conflict of interest.
