## [Peer Review File · Nature Communications]

In situ Synthesis and Dynamic Simulation of Molecularly Imprinted Polymeric Nanoparticles on a Micro-Reactor SystemReviewers' Comments:

Reviewer #1:

Remarks to the Author:

The manuscript by Erdem et al. described use of a spiral microfluidic reactor to synthesize protein imprinted polymer nanoparticles. The authors claimed several advantages of using their microreactor for continuous production of imprinted nanoparticles. The authors also used computational chemistry methods to explain possible interactions that underline the binding between the template protein and the monomer/polymer used in the experiments. The present manuscript is unfortunately lacking the novelty and impact needed for publication in Nature Communications.

My main comments:

The title should be reconsidered. The present title suggests a focus on "Spiral Micromixers". Very little was discussed on the design of the channel structure. The only exception is the length.

The productivity of the flow reactor seems to be overestimated. What will happen if the authors shorten the polymerization time in the conventional batch reactor? Does it mean no polymer particle can be obtained? Microflow reactor provides better heat exchange, safety et al. could be true, but not typically "high-throughput".

The part of molecular docking and molecular dynamics are the least relevant to the present title.

Minor comments:

Page 7, line 3 "the ultra-centrifugation step": this step is also required when the spiral flow reactor is used for synthesis, isn't it?

Page 7, line 17: "carboxylic acid groups" is incorrect, it should be "carbonyl groups" or "amide groups".

Page 12, line 12: "80 uL of BSA" is erroneous. Were you talking about protein solution?

Page 13, line 12 from bottom "A polymer mixture and initiator...over this issue": why treatment like this can solve the problem of blockage?

Reviewer #2:

Remarks to the Author:

I think that MIPs using bulk polymerization can form fixed particles within 30 min, but the reason why they usually need to be incubated for 24 hours is that the MIPs is expected to be more consistent with BSA about the structure and size. The equipment in the paper is innovative to some extent, but I see that the effect of the experiment is not particularly good. I suggest the author should increase the polymer of bulk polymerization for 30 min to compare with this experiment, rather than just select the MIPs of traditional polymerization for 24 h. How many grams of MIPs can be prepared at one time in the study? A large number of MIPs can be prepared at one time by traditional bulk polymerization.

2.1.1. "the average sizes of MIP and NIP nanoparticles were measured as 98.04 ± 0.16 nm and 161.2 ± 0.19 nm". Why are the particle sizes of MIPs and NIPs so different? The preparation process of MIPs and NIPs is almost the same.

2.3 This study lacks selectivity and interference for BSA. The structural analogues of BSA should be

selected to compete with the target for identification by MIPs at the same time. Imprinting factor is an important indicator of MIPs. Please add.
4.2 How to wash template molecules in addition to microfluidic polymerization?

Supporting Data

1.2 "Once supernatant of these nanoparticles was used for further size and concentration analysis, the imprinted nanoparticles were washed with ethanol and water before desorption and further characterization studies". Why choose supernatant? I think MIPs are insoluble.

"the imprinted nanoparticles were washed with ethanol and water before desorption"

Why elute with ethanol and water? What is the purpose of ethanol? Will it cause changes in BSA?

"The desorption of BSA protein from nanoparticles was carried out using 0.5 M NaCl"

How long does this process last?

1.6 Adsorption-desorption studies

"For this purpose, non-desorbed nanoparticles and 0.5 M NaCl solution (1:1) was shaken for 1 h at room temperature."

"Pellet was resuspended in water and then incubated with 1.5 mg/mL BSA for 1 h". Why does it take so long to adsorb? Whether the target protein will denature during this process

Generally, I think adsorption is easier than elution. Why does adsorption and elution take the same time?

COMMENTS TO AUTHORS

REVIEWER #1

Comment: The manuscript by Erdem et al. described use of a spiral microfluidic reactor to synthesize protein imprinted polymer nanoparticles. The authors claimed several advantages of using their microreactor for continuous production of imprinted nanoparticles. The authors also used computational chemistry methods to explain possible interactions that underline the binding between the template protein and the monomer/polymer used in the experiments. The present manuscript is unfortunately lacking the novelty and impact needed for publication in Nature Communications.

Response: We thank the Reviewer-1 for detailing the major highlighting outcomes that we demonstrated in the manuscript. Per the comment, we have now improved our texts in the revised manuscript by clearly stating the novelty and impact of our study. For the other comments, please find our point-by-point responses below.

We have addressed the novelty and the impact of our study in the following section:

“Abstract

Current practices in synthesizing molecularly imprinted polymers (MIPs) face challenges—lengthy process, low-productivity, the need for expensive and sophisticated equipment, and they cannot be controlled in situ synthesis. Herein, we present a micro-reactor for in situ and continuously synthesizing trillions of MIP nanoparticles that contains molecular fingerprints of bovine serum albumin in a short time (5-30 min). Initially, we performed COMSOL simulation to analyze mixing efficiency with altering flow rates, and experimentally validated the platform for synthesizing nanoparticles with sizes ranging from 52-106 nm. Molecular interactions between monomers and protein were also examined by molecular docking and dynamics simulations. Afterwards, we benchmarked the micro-reactor parameters through dispersity and concentration of MIPs using principal component analysis. Sensing assets of MIPs were examined on a metamaterial sensor, resulting in 81% of precision with high selectivity (4.5 times), and three cycles of consecutive use. Overall, our micro-reactor stood out as a superior approach owing to high-productivity (48-288 times improvement in assay-time and 2 times improvement in reagent volume), enabling to produce 1.4 - 1.5 times more MIPs, one-single step, and continuous production compared to conventional strategy.”

“1. Introduction

To our best knowledge, we present, for the first time, a micro-reactor to in situ synthesize protein-imprinted nanoparticles through a continuous flow strategy. This system was strategized through COMSOL Multiphysics simulations to analyze mixing

efficiency with different flow rates for mixtures and experimentally validated for synthesizing nanoparticles down to nm-scales (**Figure 1**). In addition, molecular docking method was employed to predict the preferred orientations when a ligand and MIPs were reacted to form a stable complex. The micro-reactor led to the synthesis of MIPs in a relatively short time (down to 5-30 min) compared to conventional batch production (24 h) for synthesizing the intended size of nanoparticles. As a model system, bovine serum albumin (BSA) was interacted with methacrylic acid as a functional monomer. Polymer solution and initiator reagent were introduced to the micro-reactor system with varying flow rates, and principal component analysis (PCA) was employed to verify the optimum conditions. The continuously synthesized nanoparticles were collected as soon as 5 min of the process (288 times more efficient compared to the conventional method), and characterized by dynamic light scattering (DLS) and nanoparticle tracking analysis (NTA). Hence, we comprehensively benchmarked our system with various parameters such as channel length, flow rate, turnaround time, binding capability, and selectivity along with a detailed comparison with a conventional batch productions strategy. Considering the translation of imprinted nanoparticles into the market, the micro-reactor system minimizes the overall cost, as well as offers multiple time use and higher yield compared to the conventional method.” (Pages 4-5)

“3. Conclusion

In this study, we developed, for the first time, a micro-reactor for the continuous synthesis of imprinted nanoparticles. Initially, we performed COMSOL simulation, and further experimentally analyzed a number of flow rates for the initiator (20 and 60 $\mu\text{L}/\text{min}$) to synthesize MIPs at these conditions. The distribution of nanoparticle sizes was found to be consistently larger at different time intervals as compared to the smaller sizes observed at lower flow rates. Although the initiator flow rate of 60 $\mu\text{L}/\text{min}$ formed nanoparticles with sequential size increments, it was unable to produce smaller-sized nanoparticles. Considering the higher surface area, we successfully produced small nanoparticles down to 52-106 nm at the initiator flow rate of 2.5 $\mu\text{L}/\text{min}$.

In addition, the presented system enables the collection of imprinted nanoparticles within 5 - 30 min; and minimizes the overall cost down to \$10 per chip, which can also be used multiple times. Besides that, in every 30 min of time intervals, the micro-reactor system provided 1.4 - 1.5 times more MIPs compared to the conventional method according to the calculated dry weights. On the other hand, the counterpart strategy—conventional method, required 24 h to produce the intended size of imprinted nanoparticles (~ 100 nm). In addition, the use of the micro-reactor is very facile for proper operation including: (i) filling two syringes with samples (one for polymer mixture and the other for initiator) and (ii) running the samples under a controlled flow rate on a pump. For this scheme of operation, a standard syringe pump is the only requirement that would be further miniaturized with existing open-source systems with a total cost

under \$100 (i.e., Bartels mp6 micropumps). Considering easy adaptation of this strategy to produce MIPs in an efficient manner, pressure/vacuum controllers and generators, as supplied, for instance, by ElveFlow®, would be a one-time purchase, and the micro-reactor would provide more controllable process in producing more complex synthesis in situ.” (Pages 16-17)

Comments

Comment-1: The title should be reconsidered. The present title suggests a focus on “Spiral Micromixers”. Very little was discussed on the design of the channel structure. The only exception is the length.

Response: According to the reviewer’s comment, we have now updated the title as **“In situ Synthesis and Dynamic Simulation of Molecularly Imprinted Polymeric Nanoparticles on a Micro-Reactor System”**. Now we believe this title represent all points achieved in the manuscript including microreactor design, MIP synthesis, and molecular docking.

In addition, we have further characterized the micro-reactor platform through COMSOL flow simulations that analyze different flow rates and channel lengths in order to validate the system from the flow dynamics perspective. The simulation results can also be found below:

Materials and Methods

“4.2. COMSOL Simulation

*In this study, COMSOL Multiphysics 5.6 was utilized to analyze the mixing efficiency with different flow rates for monomer mixture and initiator through the integration and solution of three physics, i.e., laminar flow, transport of diluted species, and chemistry. A controlled flow rate was initially established to attain stabilized conditions for the fluid flow. Upon establishing the stabilized conditions, the mixing location within the channel was analyzed, and we observed that the downstream length at which mixing occurred remained constant over time. As a result, it was established that the mixing location was independent of time, and the stationary mode was selected accordingly. A computational method was then utilized to systematically solve for three physical processes involved: laminar flow, chemistry, and transport of diluted species. Herein, we have initially run our analysis using a 2D model, which had same thickness and width values of the micro-reactor, and differed in the length value only (250 mm) in order to depict more resolution in the figures. This model precisely represented the micro-reactor designed in our study. In addition, we have also evaluated the real geometries used in our experiments including exact length (**Supplementary Materials and Methods**).” (Pages 19)*

2. RESULTS AND DISCUSSION

2.1. COMSOL simulation results

The implementation of micro-reactors enables to produce nanoparticles with adjustable physical and chemical properties, such as size, size distribution, and shape, resulting in uniform outcomes. In this regard, efficient mixing and polymerization through a chemical reaction were considered as main criteria to acquire such properties. Initially, considering the physical assets, the application of micro-reactors provides control over mixing parameters such as flow rate and ratio, resulting in efficient mixing. The way in which particles are mixed would also influence potentially both their structure and size distribution¹. In addition to efficient mixing, the geometry of the micro-reactor would have impact on the polymerization process². The design of microfluidic systems is highly influenced by the Reynolds number (Re) and the Péclet number (Pe), which enable precise control over the nanoparticle production^{3,4}. On the other hand, while evaluating the quality of nanoparticles produced on a micro-reactor, we cannot only consider efficient mixing and geometry characteristics since chemical reaction and duration time in the micro-reactor are the leading factors for the formation of nanoparticles. Accordingly, we considered simulation studies to fulfill the criteria of efficient mixing and geometry, and also, we evaluated the size and size distribution characteristics of nanoparticles as a quality criterion after performing experiments at the simulated conditions.

In this scenario, a computational method was utilized to systematically solve for three physical processes involved, i.e., laminar flow, chemistry, and transport of diluted species. As a representative purpose, we kept all the channel parameters same except the length as 250 mm at each flow condition, thereby increasing the resolution in the simulation results (**Figure 2**). We also run the simulations for the micro-reactors with 1 m, 2 m, and 3 m of length (**Figure S1**) in order to mimic the same conditions in the experiments. Briefly, each process was solved step-by-step, with the solution of each step serving as the input for the subsequent step. **Figure 2A** shows the image of microchannels with two different fluids passing through. **Figure 2B-C** depicts the inlets for the monomer mixture and initiator, along with a corresponding color bar that illustrates the mixing efficiency of two fluids. The green color on the color bar represents a higher level of mixing efficiency. Next, in the chemistry module, chemical species for the monomer mixture and initiator were defined, along with their respective molar concentrations. The diffusion coefficients were calculated within the chemistry module through an automated process. This information was then utilized in the transport of diluted species to display the total concentration of these chemical species and their level of mixing efficiency. **Figure 2D-F** presents the relationship between various flow rates and the resulting mixing efficiency. As demonstrated, the closer the flow ratio was, the higher the mixing efficiency become. The flow rate of the monomer mixture was kept constant, while the flow rate of the initiator was gradually increased,

resulting in an improved mixing efficiency observed in the comparison case of 143.75 vs 60 $\mu\text{L}/\text{min}$.

Per this observation, we also synthesized the MIPs according to the flow rates given in the simulation. According to our DLS data (**Figure 2G-I**), the synthesized nanoparticles had a size range of 52-106 nm, 170-245 nm, and 85-175 nm for the initiator flow rates of 2.5, 20, and 60 $\mu\text{L}/\text{min}$ (while keeping the flow rate for the monomer mixture same), respectively. On the other hand, the average PDI values were 0.25, 0.295, and 0.182, respectively. Considering the PDI values, all the flow rates provided similar results; but the initiator flow rate of 60 $\mu\text{L}/\text{min}$ resulted in more uniform structures. However, the size distribution of nanoparticles at different durations were mainly larger compared to the lower flow rates. Although the initiator flow rate of 60 $\mu\text{L}/\text{min}$ led to more sequential increments in the nanoparticle size that stated more controllable production, this flow rate was not able to harvest lower diameters. This might have occurred since higher flow rates of initiator were not able to interact with monomer mixture for enough time in order to produce smaller particles. Another consideration would be that higher flow rates resulted in agglomeration and sticking nanoparticles, thereby increasing their sizes. Furthermore, the initiator flow rate of 20 $\mu\text{L}/\text{min}$ enabled to produce larger sizes of nanoparticles (up to 245 nm), and it was not possible to produce at lower diameters (only down to 170 nm). The lowest flow rate of initiator (2.5 $\mu\text{L}/\text{min}$), on the other hand, resulted in yielding smaller diameters of nanoparticles (down to 52 nm) at the duration 5-20 min, and at the same time, enlarging the nanoparticle size was also possible at this flow rate while increasing the duration. Consequently, we proceeded our experiments using 1X flow rate conditions (the monomer mix flow rate of 143.75 $\mu\text{L}/\text{min}$ and initiator flow rate of 2.5 $\mu\text{L}/\text{min}$) since smaller sizes of nanoparticles would have more surface area, thereby potentially interacting more numbers of target molecules on the course of sensing studies. For this flow rate, synthesis time was also further evaluated in the following sections (Pages 5-7).

Figure 2. Flow simulations of the micro-reactor and experimental data of the synthesized MIPs. (A) The images of a micro-reactor with two different fluids (rhodamine B as red and commercial textile dye as light blue) passing through microchannels are represented. A schematic representation of micro-reactor demonstrates (B) surface concentration and (C) velocity magnitude profile, in which blue and red colors represent the monomer mix and the initiator, respectively. COMSOL simulation results state the surface concentration of MIP synthesis using the combinations of flow rates that include (D) 143.75 $\mu\text{L}/\text{min}$ and 2.5 $\mu\text{L}/\text{min}$; (E) 143.75 $\mu\text{L}/\text{min}$ and 20 $\mu\text{L}/\text{min}$; (F) 143.75 $\mu\text{L}/\text{min}$ and 60 $\mu\text{L}/\text{min}$. DLS data is presented for the combinations of flow rates, i.e., (G) 143.75 $\mu\text{L}/\text{min}$ and 2.5 $\mu\text{L}/\text{min}$; (H) 143.75 $\mu\text{L}/\text{min}$ and 20 $\mu\text{L}/\text{min}$; (I) 143.75 $\mu\text{L}/\text{min}$ and 60 $\mu\text{L}/\text{min}$. The data is evaluated with one-way ANOVA (Freidman test) statistical analysis, and the statistical difference is shown as * $p < 0.05$ and ** $p < 0.01$.

Figure S1. COMSOL simulations for mixing efficiency. The combinations of flow rates applied in the micro-reactors, different lengths of (A-C) 1 m, (D-F) 2 m, and (G-I) 3 m are evaluated. The combinations of flow rates of (A, D, G) monomer mix include 143.75 $\mu\text{L}/\text{min}$, initiator mix: 2.5 $\mu\text{L}/\text{min}$; (B, E, H) monomer mix: 143.75 $\mu\text{L}/\text{min}$, initiator mix: 20 $\mu\text{L}/\text{min}$; and (C, F, I) monomer mix: 143.75 $\mu\text{L}/\text{min}$, initiator mix: 60 $\mu\text{L}/\text{min}$.

Comment-2: The productivity of the flow reactor seems to be overestimated. What will happen if the authors shorten the polymerization time in the conventional batch reactor? Does it mean no polymer particle can be obtained? Microflow reactor provides better heat exchange, safety et al. could be true, but not typically “high-throughput”.

Response: As the reviewer’s suggestion, we have now shortened the polymerization time in both production methods (conventional method and micro-reactor) and compared them in terms of physicochemical properties. We have included these findings into the manuscript and supporting data files. The included text can be found below:

Manuscript file

“3.2. Micro-reactor synthesis of BSA-imprinted nanoparticles

“... Additionally, in order to test shorter polymerization time, we also collected MIPs at the shorter time periods (5, 10, and 20 min) for both conventional method and micro-

reactor system. The collective dry-weight of MIPs per mL collected between 30-180 min intervals were also calculated after the MIPs were lyophilized.” (Page 20)

2. Result and discussion

2.2. Characterization of BSA-imprinted nanoparticle

2.2.2. Physicochemical analysis

“We have also tested shorter polymerization times in both conventional method and micro-reactor production of MIPs. According to the results shown in **Figure S4**, conventional synthesis led to the formation of larger size of MIPs (within the range of 150-200 nm), while MIPs collected from the micro-reactor system had smaller diameters (between 52 and 106 nm). In the micro-reactor, the interactions between the initiator and the monomer mixture might be higher due to the high surface-area-to-volume ratio, which resulted in the synthesis of smaller MIPs. The collective dry-weight of MIPs were also calculated (**Table S3**), and at all the durations, the micro-reactor system provided 1.4 - 1.5 times more MIPs compared to the conventional method, and with our platform, we were able to produce 263.8 mg of MIPs within 180 min.” (Page 12)

Figure S4. DLS data and images of BSA-imprinted nanoparticles collected at time intervals varying from 0 to 180 min. DLS results are demonstrated for (A) micro-reactor and (B) conventional synthesized MIPs that are collected at different time intervals. (C) MIPs collected from micro-reactor and conventional method are compared after 5, 30, and 180 min. The photos of MIPs solutions synthesized on (D) micro-reactor and (E) conventional method are exhibited at these time slots.

Table S3. The cumulative dry-weight of MIPs collected at different time intervals.

Time (min)	Weight (mg)	
	Micro-reactor	Conventional
30	47.2	31.2
60	97.6	61.7
90	138.6	92.3
120	186.7	124.1
150	224.8	155.5
180	263.8	187

Supporting data file

“1.2. Conventional synthesis of BSA-imprinted nanoparticles

Addition to these processes, we have tested the conventional production in 5, 10, 20, 30, 60, 90, 120, 150 and 180 min to compare them with the production results derived from the micro-reactor system. The collective dry-weight of MIPs per mL collected between 30-180 min intervals were also calculated after the MIPs were lyophilized (Page 4)

Comment-3: The part of molecular docking and molecular dynamics are the least relevant to the present title.

Response: We believe studies regarding to molecular docking and molecular dynamics are essential to explore how molecules interacts with each other. In particular, the literature has been missing such information and analysis for the MIP synthesis. We agree with the reviewer’s comment, and for this reason, we have now included molecular docking to our title. Our updated title is **“In situ Synthesis and Dynamic Simulation of Molecularly Imprinted Polymeric Nanoparticles on a Micro-Reactor System”**.

Minor comments:

Comment-4: Page 7, line 3 “the ultra-centrifugation step”: this step is also required when the spiral flow reactor is used for synthesis, isn't it?

Response: In conventional synthesis of MIPs, “sequential” centrifugation steps are needed to decrease the size of the MIPs (⁵⁻⁷). On the other hand, during the micro-reactor synthesis of MIPs, we centrifuged nanoparticles “only once” to separate MIPs from non-polymerizing materials and large particles. We have now updated the regarding section as follows:

“...We further compared the workflow procedure of micro-reactor system with currently employed conventional method. For instance, as reported in the literature⁵⁻⁹, the sequential centrifugation steps (4-5 times, and at the speeds of 14.500-30.000 rpm) are required in the conventional method to decrease the nanoparticle size. On the other hand, using the micro-reactor system, we centrifuged nanoparticles only once at low speeds (6000 rpm) to separate them from non-polymerizing materials and large particles. MIPs synthesized with the conventional method might have possess polydisperse properties due to reactions in a bulk environment. The purpose of sequentially repeated centrifuges is to separate particles of different sizes and to obtain the smallest particles in a monodisperse form. However, multi-step method is costly in terms of time and effort. With the micro-reactor system, we were able to synthesize particles within nm-scale with monodisperse assets without requiring the repeated centrifugation steps (only one single step of centrifugation) that decrease the process time and eventually increase cost effectiveness of the process.” (Pages 11-12)

Comment-5: Page 7, line 17: “carboxylic acid groups” is incorrect, it should be “carbonyl groups” or “amide groups”.

Response: We thank to the reviewer for the attentive comment. We have now corrected the statement as follows:

“When BSA protein was desorbed from MIPs, the intensity of peaks corresponding to C=O bonding decreased due to the reduction of carbonyl groups originated from BSA.” (Page 13)

Comment-6: Page 12, line 12: “80 uL of BSA” is erroneous. Were you talking about protein solution?

Response: We have now corrected the statement as follows:

“... In the present study, we only run 20 μ L of MA solution (monomer), 0.25 mL of HEMA solution (co-monomer), 0.5 mL of EDGMA solution, and 80 μ L of BSA solution (in pH 4 buffer) as a model target, whilst the conventional method required 40 μ L of

MA solution (monomer), 0.5 mL of HEMA solution (co-monomer), 1 mL of EDGMA solution, and 160 μ L of BSA solution, hence minimizing the volume of reagents down to 50%.” (Page 17)

Comment-7: Page 13, line 12 from bottom “A polymer mixture and initiator...over this issue”: why treatment like this can solve the problem of blockage?

Response: We have now clarified the solution for the potential blockage problem and updated the text accordingly.

“Polymerization is highly dependent on the concentration of the initiator and the monomer mixture. These values should be defined accordingly to avoid blockage problem in the microchannels, as they may directly impact the polymerization rate in the microreactor. In our study, we have considered the values used in the batch process and adapted them to our micro-reactor system. The critical point is that both solutions should simultaneously be introduced to the micro-reactor. This way, more uniform polymerization could be provided potentially. However, due to the different flow rates of both solutions (2.5 μ L/min and 143.75 μ L/min), it was highly challenging to introduce them into the system at the same time. When the initiator was introduced into the micro-reactor before the monomer mixture, we observed some accumulations of initiator in the initial part of the main channel, which resulted in intense polymerization upon meeting the monomer mixture. The critical point is that both solutions should be introduced simultaneously to the micro-reactor. This way, more uniform polymerization could be provided potentially.” (Pages 17-18)

REVIEWER #2

Comment: I think that MIPs using bulk polymerization can form fixed particles within 30 min, but the reason why they usually need to be incubated for 24 hours is that the MIPs is expected to be more consistent with BSA about the structure and size. The equipment in the paper is innovative to some extent, but I see that the effect of the experiment is not particularly good. I suggest the author should increase the polymer of bulk polymerization for 30 min to compare with this experiment, rather than just select the MIPs of traditional polymerization for 24 h. How many grams of MIPs can be prepared at one time in the study? A large number of MIPs can be prepared at one time by traditional bulk polymerization.

Response: We thank the Reviewer-2 for highlighting the innovative assets of our study and manuscript, and we have now included more details on how we improve the current state-of-art through the presented micro-reactor system. Therefore, we updated our novelty statements in the manuscript. In addition, we presented more collection time slots spanning from 5 min to 180 min for both micro-reactor system and conventional method. We also calculated and compared the mass of the produced MIPs at these methods.

Novelty: Please find our updates on the novelty and the impact of our study below:

“Abstract

Current practices in synthesizing molecularly imprinted polymers (MIPs) face challenges—lengthy process, low-productivity, the need for expensive and sophisticated equipment, and they cannot be controlled in situ synthesis. Herein, we present a micro-reactor for in situ and continuously synthesizing trillions of MIP nanoparticles that contains molecular fingerprints of bovine serum albumin in a short time (5-30 min). Initially, we performed COMSOL simulation to analyze mixing efficiency with altering flow rates, and experimentally validated the platform for synthesizing nanoparticles with sizes ranging from 52-106 nm. Molecular interactions between monomers and protein were also examined by molecular docking and dynamics simulations. Afterwards, we benchmarked the micro-reactor parameters through dispersity and concentration of MIPs using principal component analysis. Sensing assets of MIPs were examined on a metamaterial sensor, resulting in 81% of precision with high selectivity (4.5 times), and three cycles of consecutive use. Overall, our micro-reactor stood out as a superior approach owing to high-productivity (48-288 times improvement in assay-time and 2 times improvement in reagent volume), enabling to produce 1.4 - 1.5 times more MIPs, one-single step, and continuous production compared to conventional strategy.”

“1. Introduction

To our best knowledge, we present, for the first time, a micro-reactor to in situ synthesize protein-imprinted nanoparticles through a continuous flow strategy. This system was strategized through COMSOL Multiphysics simulations to analyze mixing efficiency with different flow rates for mixtures and experimentally validated for synthesizing nanoparticles down to nm-scales (Figure 1). In addition, molecular docking method was employed to predict the preferred orientations when a ligand and MIPs were reacted to form a stable complex. The micro-reactor led to the synthesis of MIPs in a relatively short time (down to 5-30 min) compared to conventional batch production (24 h) for synthesizing the intended size of nanoparticles. As a model system, bovine serum albumin (BSA) was interacted with methacrylic acid as a functional monomer. Polymer solution and initiator reagent were introduced to the micro-reactor system with varying flow rates, and principal component analysis (PCA) was employed to verify the optimum conditions. The continuously synthesized nanoparticles were collected as soon as 5 min of the process (288 times more efficient compared to the conventional method), and characterized by dynamic light scattering (DLS) and nanoparticle tracking analysis (NTA). Hence, we comprehensively benchmarked our system with various parameters such as channel length, flow rate, turnaround time, binding capability, and selectivity along with a detailed comparison with a conventional batch productions strategy. Considering the translation of imprinted nanoparticles into the market, the micro-reactor system minimizes the overall cost, as well as offers multiple time use and higher yield compared to the conventional method.” (Pages 4-5)

“3. Conclusion

In this study, we developed, for the first time, a micro-reactor for the continuous synthesis of imprinted nanoparticles. Initially, we performed COMSOL simulation, and further experimentally analyzed a number of flow rates for the initiator (20 and 60 $\mu\text{L}/\text{min}$) to synthesize MIPs at these conditions. The distribution of nanoparticle sizes was found to be consistently larger at different time intervals as compared to the smaller sizes observed at lower flow rates. Although the initiator flow rate of 60 $\mu\text{L}/\text{min}$ formed nanoparticles with sequential size increments, it was unable to produce smaller-sized nanoparticles. Considering the higher surface area, we successfully produced small nanoparticles down to 52-106 nm at the initiator flow rate of 2.5 $\mu\text{L}/\text{min}$.

In addition, the presented system enables the collection of imprinted nanoparticles within 5 - 30 min; and minimizes the overall cost down to \$10 per chip, which can also be used multiple times. Besides that, in every 30 min of time intervals, the micro-reactor system provided 1.4 - 1.5 times more MIPs compared to the conventional method according to the calculated dry weights. On the other hand, the counterpart strategy—conventional method, required 24 h to produce the intended size of imprinted

nanoparticles (~100 nm). In addition, the use of the micro-reactor is very facile for proper operation including: (i) filling two syringes with samples (one for polymer mixture and the other for initiator) and (ii) running the samples under a controlled flow rate on a pump. For this scheme of operation, a standard syringe pump is the only requirement that would be further miniaturized with existing open-source systems with a total cost under \$100 (i.e., Bartels mp6 micropumps). Considering easy adaptation of this strategy to produce MIPs in an efficient manner, pressure/vacuum controllers and generators, as supplied, for instance, by ElveFlow®, would be a one-time purchase, and the micro-reactor would provide more controllable process in producing more complex synthesis *in situ*.” (Pages 16-17)

Polymerization time and mass measurements: According to the reviewer’s comment, we have now shortened the polymerization time in both production methods (conventional method and micro-reactor system) and compared them in terms of physicochemical properties. We have also collected and freeze-dried the MIPs at different time intervals of both methods. Per the results, conventional method produced larger sizes of MIPs while we achieved to produce smaller sizes of MIPs in the micro-reactor system. In addition, the collective dry-weight of MIPs were also calculated. At all durations, the micro-reactor system provided 1.4 - 1.5 times more MIPs compared to the conventional method, and with our platform, we were able to produce 263.8 mg of MIPs within 180 min (**Table S3**). Please see these updates on the manuscript and supporting information files as follows:

Manuscript file

“3.2. Micro-reactor synthesis of BSA-imprinted nanoparticles

“... Additionally, in order to test shorter polymerization time, we also collected MIPs at the shorter time periods (5, 10, and 20 min) for both conventional method and micro-reactor system. The collective dry-weight of MIPs per mL collected between 30-180 min intervals were also calculated after the MIPs were lyophilized.” (Page 20)

2. Result and discussion

2.2. Characterization of BSA-imprinted nanoparticle

2.2.2. Physicochemical analysis

“We have also tested shorter polymerization times in both conventional method and micro-reactor production of MIPs. According to the results shown in **Figure S4**, conventional synthesis led to the formation of larger size of MIPs (within the range of 150-200 nm), while MIPs collected from the micro-reactor system had smaller diameters (between 52 and 106 nm). In the micro-reactor, the interactions between the initiator and the monomer mixture might be higher due to the high surface-area-to-

volume ratio, which resulted in the synthesis of smaller MIPs. The collective dry-weight of MIPs were also calculated (**Table S3**), and at all the durations, the micro-reactor system provided 1.4 - 1.5 times more MIPs compared to the conventional method, and with our platform, we were able to produce 263.8 mg of MIPs within 180 min.” (Page 12)

Figure S4. DLS data and images of BSA-imprinted nanoparticles collected at time intervals varying from 0 to 180 min. DLS results are demonstrated for (A) micro-reactor and (B) conventional synthesized MIPs that are collected at different time intervals. (C) MIPs collected from micro-reactor and conventional method are compared after 5, 30, and 180 min. The photos of MIPs solutions synthesized on (D) micro-reactor and (E) conventional method are exhibited at these time slots.

Table S3. The cumulative dry-weight of collected MIPs at different time intervals

Time (min)	Weight (mg)	Weight (mg)
	Micro-reactor	Conventional
30	47.2	31.2
60	97.6	61.7
90	138.6	92.3
120	186.7	124.1

150	224.8	155.5
180	263.8	187

Supporting data file

“1.2. Conventional synthesis of BSA-imprinted nanoparticles

“Addition to these processes, we have tested the conventional production in 5, 10, 20, 30, 60, 90, 120, 150 and 180 min to compare them with the production results derived from the micro-reactor system. The collective dry-weight of MIPs per mL collected between 30-180 min intervals were also calculated after the MIPs were lyophilized.” (Page 4)

Comments

Comment-1: 2.1.1.” the average sizes of MIP and NIP nanoparticles were measured as 98.04 ± 0.16 nm and 161.2 ± 0.19 nm”. Why are the particle sizes of MIPs and NIPs so different? The preparation process of MIPs and NIPs is almost the same.

Response: We have now provided a detailed explanation of the potential reasons for larger sizes of NIP nanoparticles. Please see the revised text below:

“2.2.1. Physicochemical analysis

“...Consequently, MIP nanoparticles are anticipated to have a more consistent size and shape, in contrast to NIP nanoparticles. As we and the others observed in different synthesis methods^{10,11}, the size of MIP nanoparticles is typically smaller than those of NIP nanoparticles since the template molecule employed in the molecular imprinting process is able to possibly limit the size of the produced nanoparticles. In principle, functional monomers can form hydrogen-bonded dimers in the absence of a template, while synthesizing NIP nanoparticles, and the pre-polymerization solution contains both functional monomer dimers and free functional monomers. There are other possible molecular interactions between functional monomer and template in the MIPs that may potentially alter the formation of cross-linked polymer nuclei¹². Although the preparation and polymerization processes were almost the same for synthesizing MIPs and NIPs, NIPs contained more functional monomer dimers in the pre-polymerization solution and the numbers of polymerization nuclei were generally less than in those of MIPs. This might have led to the size of the growing final cross-linked NIP nuclei to be significantly larger than that of the MIP nanoparticles. Moreover, the paucity of specific

cavities on NIPs could have potentially caused them to have larger dimensions than MIPs. Since only the target molecule that fits inside the cavity were kept and the remaining one was washed away, the selective binding in the MIPs concluded to a decrease in the size of the nanoparticles¹³.” (Pages 9-10)

Comment-2: 2.3 This study lacks selectivity and interference for BSA. The structural analogues of BSA should be selected to compete with the target for identification by MIPs at the same time. Imprinting factor is an important indicator of MIPs. Please add.

Response: As described by the reviewer, selectivity analysis compares the binding effect of structurally analogue molecules with the target molecules. On the other hand, imprinting factor (relative selectivity (k')) defines the effect of molecular grooves for the binding of target molecules, thereby comparing the binding impact of MIPs and NIPs.

We have already performed selectivity analysis with Human Serum Albumin (HSA) protein due to molecular similarities in the chemical composition. According to the results, the selectivity coefficient was calculated as 4, and BSA-imprinted MIP nanoparticles showed higher selectivity than HSA protein. Per the comment, we have now calculated the imprinting factor as follows:

Manuscript

“2.6. Selectivity, imprinting factor, and repeatability analysis

*Briefly, selectivity analysis compares the binding effects of structurally analogue molecules (HSA) with the target molecules (BSA). Hence, selectivity performance of imprinted nanoparticles synthesized on the micro-reactor system was evaluated while testing their binding with HSA protein (20 μ M) due to molecular similarity in the chemical compositions. As shown in **Figure 7G**, a wavelength shift of 0.9 ± 0.1 nm occurred when BSA protein (20 μ M) was introduced into the system, while this value was 0.2 ± 0.1 nm for HSA (20 μ M). The selectivity (k) coefficient was calculated according to **Equation 1** as follows:*

$$k = \text{Shift}_{BSA} / \text{Shift}_{HSA}$$

Equation 1

Considering the earlier reports¹⁴, we calculated the selectivity coefficient (k) as 4.5. If k value is greater than 1, it demonstrates high affinity of MIPs to the target molecule, i.e., BSA protein⁵.

In addition, imprinting factor (relative selectivity (k')) defines the effect of molecular grooves for the binding of target molecules, thereby comparing the binding impact of MIPs and NIPs. The relative selectivity (k') was calculated using the **Equation 2**.

$$k' = k_{MIP}/k_{NIP}$$

Equation 2

While applying 30 μM of BSA solution to both MIPs and NIPs, we observed that the response for the NIPs was 0.9 ± 0.3 nm, and this value was found to be 2.1 ± 0.5 nm for MIPs. Hence, the imprinted nanoparticles were able to detect BSA proteins with 2.3 times greater than the NIP nanoparticles (**Figure 7H**)." (Pages 14-15)

Figure 7. Binding studies. (A-F) The plots display the binding results of BSA protein solutions at different concentrations (A): 10 μM , (B) 20 μM , (C) 30 μM , (D) 40 μM , and (E) 50 μM . (F) The calibration curve is plotted for evaluating the plasmonic signals upon the increments of protein concentrations. (G) As a selectivity analysis, the binding of BSA and HSA proteins is evaluated on the sensor modified with MIP nanoparticles. Wavelength shifts are collected in real-time data for BSA and HSA solutions at the same concentrations (20 μM for both protein solutions), and this plot states the selective binding of BSA protein to the MIPs. (H) Imprinting factor is evaluated through the real-time responses collected for the binding of BSA (30 μM) to MIPs and NIPs. (I) Three times adsorption and desorption cycles are evaluated by introducing 30 μM of BSA and 0.5 M of NaCl as desorption buffer.

Supporting data

“1.10. Selectivity, imprinting factor, and repeatability studies

...The selectivity of BSA-imprinted nanoparticles was also examined. We herein applied 20 μ M of human serum albumin (HSA) solution onto the plasmonic sensor, which was modified with BSA-imprinted nanoparticles, and we also compared the results derived from the same concentration of BSA applied to the same sensor. According to the results, the selectivity was calculated. In addition, we calculated imprinting factor (relative selectivity (k')) by experimentally analyzing the binding of BSA proteins to the MIPs and NIPs.” (Page 7)

Comment-3: 4.2 How to wash template molecules in addition to microfluidic polymerization?

Response: We have now updated the statement. Please see it below:

“4.3. Micro-reactor synthesis of BSA-imprinted nanoparticles

“...The desorption of BSA protein solution from MIP nanoparticles was carried out using 0.5 M of NaCl for an hour until no absorbance was observed at 280 nm.” (Page 20)

Comment-4: 1.2 “Once supernatant of these nanoparticles was used for further size and concentration analysis, the imprinted nanoparticles were washed with ethanol and water before desorption and further characterization studies”. Why choose supernatant? I think MIPs are insoluble. “the imprinted nanoparticles were washed with ethanol and water before desorption” Why elute with ethanol and water? What is the purpose of ethanol? Will it cause changes in BSA?

Response: After the polymerization, the nanoparticle solution was washed with ethanol and water to remove impurities (cleaning purposes only) and excess monomers, respectively. The desorption or elution was carried out with 0.5 M of NaCl. Moreover, the reason for using supernatant is to eliminate larger sizes of nanoparticles. We have now indicated this information to the revised supporting data.

“..Once the supernatant of these nanoparticles was used for further size and concentration analysis, the MIPs were washed with ethanol and water before desorption and further characterization studies. Washing with ethanol and water enables to remove impurities (for cleaning purposes only) and excess monomers, respectively. Moreover, the reason for using supernatant is to eliminate larger sizes of nanoparticles. The desorption of BSA protein from imprinted nanoparticles was carried out using 0.5 M of NaCl for an hour until no absorbance was observed at 280 nm.” (Pages 4)

Comment-5: “The desorption of BSA protein from nanoparticles was carried out using 0.5 M NaCl” How long does this process last?

Response: We have now updated the statement. Please see it below:

“The desorption of BSA protein solution from MIP nanoparticles was carried out using 0.5 M of NaCl for an hour until no absorbance was observed at 280 nm.” (Page 20)

1.6. Adsorption-desorption studies

Comment-6: “For this purpose, non-desorbed nanoparticles and 0.5 M NaCl solution (1:1) was shaken for 1 h at room temperature.”

“Pellet was resuspended in water and then incubated with 1.5 mg/mL BSA for 1 h”. Why does it take so long to adsorb? Whether the target protein will denature during this process. Generally, I think adsorption is easier than elution. Why does adsorption and elution take the same time?

Response: The desorption process was carried out for an hour to ensure that all bound target proteins (BSA) were removed from the binding sites on the MIPs as demonstrated in the literature ^{15,16}. The target protein would not denature during this process.

“...Pellet was resuspended in water and then incubated with 1.5 mg/mL BSA for an hour in order to ensure that all bound target protein (BSA) were removed from the binding sites on the MIPs as demonstrated in the literature ^{15,16}. After centrifugation step, the supernatant was removed and measured at 280 nm. During this process, the target protein would not be denatured.” (Page 6)

REFERENCES

1. Liu, D., Zhang, H., Fontana, F., Hirvonen, J. T. & Santos, H. A. Microfluidic-assisted fabrication of carriers for controlled drug delivery. *Lab Chip* **17**, 1856–1883 (2017).
2. Thiermann, R. *et al.* Size controlled polymersomes by continuous self-assembly in micromixers. *Polymer (Guildf)*. **53**, 2205–2210 (2012).
3. Hessel, V., Löwe, H. & Schönfeld, F. Micromixers—a review on passive and active mixing principles. *Chem. Eng. Sci.* **60**, 2479–2501 (2005).
4. Fabozzi, A. *et al.* Design of functional nanoparticles by microfluidic platforms as advanced drug delivery systems for cancer therapy. *Lab Chip* (2023).
5. Erdem, Ö., Saylan, Y., Cihangir, N. & Denizli, A. Molecularly imprinted nanoparticles based plasmonic sensors for real-time *Enterococcus faecalis* detection. *Biosens. Bioelectron.* **126**, 608–614 (2019).
6. Omid, F., Behbahani, M., Sadeghi Abandansari, H., Sedighi, A. & Shahtaheri, S. J. Application of molecular imprinted polymer nanoparticles as a selective solid phase extraction for preconcentration and trace determination of 2, 4-dichlorophenoxyacetic acid in the human urine and different water samples. *J. Environ. Heal. Sci. Eng.* **12**, 1–10 (2014).
7. Kempe, H. & Kempe, M. Ouzo polymerization: A bottom-up green synthesis of polymer nanoparticles by free-radical polymerization of monomers spontaneously nucleated by the Ouzo effect; Application to molecular imprinting. *J. Colloid Interface Sci.* **616**, 560–570 (2022).
8. Saylan, Y., Akgönüllü, S. & Denizli, A. Preparation of magnetic nanoparticles-assisted plasmonic biosensors with metal affinity for interferon- α detection. *Mater. Sci. Eng. B* **280**, 115687 (2022).
9. Erdem, Ö., Cihangir, N., Saylan, Y. & Denizli, A. Comparison of molecularly imprinted plasmonic nanosensor performances for bacteriophage detection. *New J. Chem.* **44**, 17654–17663 (2020).
10. Erdem, V. Z., Oktay Başeğmez, H. İ. & Baydemir Peşint, G. AFB1 recognition from liver tissue via AFB1 imprinted magnetic nanoparticles. *J. Chromatogr. B* **1210**, 123453 (2022).
11. Qin, D., Wang, J., Ge, C. & Lian, Z. Fast extraction of chloramphenicol from marine sediments by using magnetic molecularly imprinted nanoparticles. *Microchim. Acta* **186**, 428 (2019).
12. Seifi, M. *et al.* Preparation and study of tramadol imprinted micro-and nanoparticles by precipitation polymerization: Microwave irradiation and conventional heating method. *Int. J. Pharm.* **471**, 37–44 (2014).
13. Tong, Z. *et al.* Preparation and application of simetryn-imprinted nanoparticles in triazine herbicide residue analysis. *J. Sep. Sci.* **43**, 1107–1118 (2020).
14. Becskerekı, G., Horvai, G. & Tóth, B. The selectivity of molecularly imprinted polymers. *Polymers (Basel)*. **13**, (2021).
15. Çelik, O., Saylan, Y., Göktürk, I., Yılmaz, F. & Denizli, A. A surface plasmon resonance sensor with synthetic receptors decorated on graphene oxide for

- selective detection of benzylpenicillin. *Talanta* **253**, 123939 (2023).
16. Lu, W., Wang, S., Liu, R., Guan, Y. & Zhang, Y. Human serum albumin-imprinted polymers with high capacity and selectivity for abundant protein depletion. *Acta Biomater.* **126**, 249–258 (2021).

Reviewers' Comments:

Reviewer #1:

Remarks to the Author:

The authors have made significant improvement to the original manuscript. The revision has addressed most of my previous comments. Acceptance for publication is recommended.

Reviewer #3:

Remarks to the Author:

Manuscript 'In situ Synthesis and Dynamic Simulation of Molecularly Imprinted Polymeric Nanoparticles on a Micro-Reactor System'

In this research, authors assessed the formation of molecularly imprinted polymers (MIPs). Micro-reactor for in situ and continuously synthesizing of MIP nanoparticles that contains imprints of bovine serum albumin in a short time. Initially, authors performed COMSOL simulation to analyze mixing efficiency with altering flow rates, and experimentally validated the platform for synthesizing nanoparticles. Molecular interactions between monomers and proteins were also examined by molecular docking and dynamics simulations.

MIP-based electrochemical sensors are very promising for analytical and some other applications. This investigation is very interesting, from the point of view of material chemistry, and analytical chemistry. The research is in the scope of the journal. The manuscript was significantly improved after improvements recommended by some reviewers. Therefore, the manuscript can be published after minor corrections and improvements:

In introduction not well addresses protein-imprinted MIPs. Therefore, recent research on the development of electrochemical sensors based on protein-imprinted polymers (Evaluation of the interaction between SARS-CoV-2 spike glycoproteins and the molecularly imprinted polypyrrole. *Talanta* 2023, 253, 123981. // Molecularly Imprinted Polypyrrole-based Sensor for the Detection of SARS-CoV-2 Spike Glycoprotein. *Electrochimica Acta* 2022, 403, 139581. // Development of Molecularly Imprinted Polymer-based Phase Boundaries for Sensors Design (Review). *Advances in Colloids and Interface Science* 2022, 305, 102693. // Molecularly imprinted polymers for the determination of cancer biomarkers. *International Journal of Molecular Sciences* 2023, 24, 4105.) could be overviewed and discussed.

Other methods for the formation of MIPs including electrochemical could be discussed in the introduction and discussion parts of the manuscript.

The durability (assessment of the performance of the system vs time) and efficiency of the proposed system should advanced and discussed additionally.

REVIEWERS' COMMENTS

Reviewer-1 (Remarks to the Author):

Comment: The authors have made significant improvement to the original manuscript. The revision has addressed most of my previous comments. Acceptance for publication is recommended.

Response: We thank the Reviewer-1 for highlighting our significant improvements to the manuscript, and recommending for the publication in Nature Communications.

Reviewer-3 (Remarks to the Author):

Manuscript 'In situ Synthesis and Dynamic Simulation of Molecularly Imprinted Polymeric Nanoparticles on a Micro-Reactor System'

Comment: In this research, authors assessed the formation of molecularly imprinted polymers (MIPs). Micro-reactor for in situ and continuously synthesizing of MIP nanoparticles that contains imprints of bovine serum albumin in a short time. Initially, authors performed COMSOL simulation to analyze mixing efficiency with altering flow rates, and experimentally validated the platform for synthesizing nanoparticles. Molecular interactions between monomers and proteins were also examined by molecular docking and dynamics simulations. MIP-based electrochemical sensors are very promising for analytical and some other applications. This investigation is very interesting, from the point of view of material chemistry, and analytical chemistry. The research is in the scope of the journal. The manuscript was significantly improved after improvements recommended by some reviewers. Therefore, the manuscript can be published after minor corrections and improvements:

Response: We thank the Reviewer-3 for highlighting our advances and improvements in the manuscript, and also, stating the significance of our strategy and platform from the point of view of material chemistry and analytical chemistry. We have now closely followed and addressed the Reviewer-3's minor comments point-by-point as shown below.

Comment: In introduction not well addresses protein-imprinted MIPs. Therefore, recent research on the development of electrochemical sensors based on protein-imprinted polymers (Evaluation of the interaction between SARS-CoV-2 spike glycoproteins and the molecularly imprinted polypyrrole. *Talanta* 2023, 253, 123981. // Molecularly Imprinted Polypyrrole-based Sensor for the Detection of SARS-CoV-2 Spike Glycoprotein. *Electrochimica Acta* 2022, 403, 139581. // Development of Molecularly Imprinted Polymer-based Phase Boundaries for Sensors Design (Review). *Advances in Colloids and Interface Science* 2022, 305, 102693. // Molecularly imprinted polymers for the determination of cancer biomarkers. *International Journal of Molecular Sciences* 2023, 24, 4105.) could be overviewed and discussed.

Response: We have now discussed the development of electrochemical sensors based on protein-imprinted polymers in the section of Introduction.

Moreover, among biosensing modalities, we stated that MIPs are extensively utilized in electrochemical and optical detection systems. As a proof of concept, in this study, we evaluated the binding performance of MIPs synthesized on the micro-reactor using optical sensors (plasmonic metamaterial sensor). We have now indicated this information to the section of Discussion.

1. INTRODUCTION

"[...] Owing to all these advances, MIPs have been employed in a myriad of sensing applications in medicine, security, and environmental monitoring through their integrations with optical, electrochemical, and photoelectrochemical sensing¹⁻⁴. As an example, an electrochemical sensor translates interactions between an analyte and a receptor accommodating on the surface of an electrode into a quantifiable output signal (e.g., potential, current, conductivity or impedance)⁵. Viral proteins⁶⁻⁸ and protein biomarkers⁹⁻¹¹ were successfully detected on MIPs-incorporated with electrochemical sensors. However, varying

ionic strength and content in biospecimens hampers their performance. Hence, washing out the sensors with non-ionic solutions would be a necessary step to improve the signals¹² and also, the use of such materials with certain electrochemical properties would differentiate the signals from the artifacts. In this regard, especially, conducting polymers (e.g., polyaniline, poly(3,4-ethylenedioxythiophene), polypyrrole, and polythiophene) exhibit notable electrical capacitance^{13–16} or transfer electrical charges from some redox proteins and other biological entities¹⁷. They can be implemented onto the surface of electrodes, thereby forming mechanically stable layers via chemical synthesis¹⁸ electrochemical deposition¹³, enzymatic formation¹⁹, and microorganism assisted polymerization^{20–22}. Among these conducting polymers, polypyrrole has been utilized in forming MIPs for detecting low number of molecules such as proteins^{23,24}, antibiotics^{25,26}, ions^{27,28}, and so on. On the other hand, optical sensors employ light-matter coupling scheme to convert binding actions into quantifiable signals²⁹, and this strategy has been applied into the detection of this strategy has been applied into the detection of proteins, viruses, and extracellular vesicles^{30–33}. Yet, the refractive index of medium mostly dominates the optical signals while applying biospecimens. Thereby, refreshing the medium after the binding needs to be sufficient, and also, integrating anti-fouling agents would be a key step for minimizing such interfering factors.

Considering analytes, MIPs are very versatile that can be prepared in accordance with plentiful targets, and hence, they can be used for the detection of analytes, such as viruses³⁴, bacteria³⁵, various biomarkers^{36–38}, proteins^{39,40}, and chemical contaminants^{41,42}. In particular, proteins are significant disease biomarkers and essential macromolecules of organisms that involve in inter- and intracellular activities^{43–46}. MIPs have been designed in detecting protein biomarkers because of their relevance in health-related processes⁴⁷. In this manner, much emphasis has been given to the preparation of polymers for precise and selective isolation of proteins from complex biospecimens, particularly for biomedical and diagnostic applications⁴⁸. In contrast to smaller templates, proteins are complex biopolymers with a wide variety of functional groups accessible for interacting with functional monomers. Their varying regions would have significantly distinct physicochemical characteristics, such as hydrophilicity/hydrophobicity, molecular grooves, and different charges⁴⁹. Additionally, a number of protein imprinting techniques, such as boronate affinity-based molecular imprinting⁵⁰, solid-phase synthesis⁵¹, and post-imprinting modification⁵², have also been developed for the applications of separation⁵³ and purification⁵⁴, proteomics, biomarker detection³⁷, bioimaging⁵⁵, and therapy⁵⁶. ” (Pages 3-4)

3.DISCUSSION

“Among biosensing modalities, MIPs are extensively utilized in electrochemical and optical detection systems. As a proof of concept, in this study, we evaluated the binding performance of MIPs synthesized on the micro-reactor using optical sensors (plasmonic metamaterial sensor).” (Page 19)

Comment: Other methods for the formation of MIPs including electrochemical could be discussed in the introduction and discussion parts of the manuscript.

Response: In the earlier version, we stated the other methods for forming MIPs, and we have now indicated “electrochemical methods” in the section of Introduction. Moreover, we have now further discussed all these methods in the section of Discussion.

1. INTRODUCTION

“[...] The most prevalent techniques employed in MIPs synthesis include bulk^{57–59}, precipitation^{60,61}, emulsion^{62,63}, suspension^{64,65}, core–shell polymerization^{66,67}, and electrochemical methods⁶⁸. In current practice, MIP production is employed as a bulk process, and basically, their synthesis is accomplished by magnetically stirring or shaking polymeric mixture in a closed environment. Such techniques are usually time-consuming (~24 h) and labor-intensive (multiple-steps up to 10), and also require sophisticated equipment impeding their productivity.” (Page 4)

3. DISCUSSION

“[...] MIPs can be synthesized through various methods, such as bulk, precipitation, emulsion, suspension, core–shell polymerization, and electrochemical methods. Briefly, in bulk imprinting technique, target analyte is incorporated directly into the polymer matrix⁶⁹. In this conventional imprinting method, monomers and crosslinkers are polymerized in the presence of template molecules by thermal/photo initiation, and then, a bulk polymer is formed⁷⁰. However, this strategy utilizes large volume of reagents and takes lengthy processes. Another technique used for the fabrication of MIPs is emulsion polymerization, which enables the synthesis of monodispersed MIP nanoparticles containing surface-exposed binding sites. Polymerization typically takes place in the existence of a surfactant in oil-in-water emulsions⁷¹, and the monomers is commonly dissolved in an aqueous solution without the use of surfactants or emulsifiers⁷². In this method, the binding sites on the surface of imprinted microspheres or nanoparticles are evenly distributed, and the reuse rate of the MIPs obtained is high⁷³. Yet, the use of water and surfactants during polymerization between template and functional monomer can cause precipitations potentially⁷⁴. Another frequently used process is suspension polymerization. Uniform and similarly sized microspheres are obtained from small colloidal drops of the polymerization mixture suspended in a liquid phase with this method. The purpose of suspension polymerization is to obtain a homogeneous distribution of round-shaped MIP particles with sufficient binding properties⁷⁵. However, controlling particle size distribution is a challenge in suspension polymerization since the liquid droplet size distribution depends on the type and concentration of surfactant, the quality of agitation, and physical properties, such as density, interfacial tension⁷⁶. Besides, using the core-shell structure to increase MIP performance is an appealing option. The surface-imprinted shell assures that the bonding sites on the surface are evenly distributed. This analysis results in faster and more efficient template molecule removal and reattachment with enhanced binding kinetics. Such an approach provides MIPs with uniform and predetermined particle sizes. Moreover, MIP coated core-shell nanoparticles become multifunctional with the magnetic, optical or semiconducting properties of the inorganic core⁷⁷. However, there are some considerations in this process: (i) most of the interactions of macromolecules (i.e., proteins) with other molecules occurs in water and not in the organic system. (ii) Core-shell synthesis in aqueous media needs to be well investigated while applying this method for imprinting of macromolecules. (iii) More biocompatible and hydrophilic functional monomers that can reduce the effect of hydrogen bonding needs to be preferred⁷⁸. The latter, electrochemical method, is particularly employed for conducting polymers, and it relies on multiple parameters including: (i) applied voltage, (ii) potential sweep rate, (iii) the control of charge, (iv) the duration and periods of applied potential pulses, (v) any external treatment such as ultrasound, and (vi) the variations in ion and material concentrations⁶⁸. Therefore, thickness, ion permeability, density, and porosity are precisely controlled by tuning such parameters. On the other hand, overoxidization of polymers is crucial for forming MIPs, and such polymeric

structures can be applied for (1) generating oxidized radicals in order to rise sensitivity and/or selectivity towards target molecules, and (2) facilitating template removal and/or regeneration of MIP-based structures ⁶⁸. Among aforementioned conducting polymers, polypyrrole can be easily synthesized through chemical and electrochemical methods, and it benefits more from overoxidization fashion in forming MIPs. Yet, there would be more attempts to improve and broaden the applications of polypyrrole-based MIPs in the field. On the other hand, polypyrrole exhibits notable compatibility with biological entities and does not irritate immune system of mammals, thereby holding great potential to be applicable in developing implantable biosensors in the near future.” (Page 18-20)

Comment: The durability (assessment of the performance of the system vs time) and efficiency of the proposed system should advanced and discussed additionally.

Response: We have now discussed the durability of the system and the produced MIPs within a period of time, and also, elaborated the efficiency of the entire system in the section of “Discussion”.

3.DISCUSSION

“[...] The remaining challenges associated with the micro-reactor synthesis of imprinted nanoparticles and efficiency of the entire system include limitations in scale-up due to the miniaturized nature of microfluidic devices. Parallelization, which is placing many replicates micro-reactors that operate in parallel, can be proposed as a successful strategy to increase the throughput of the MIPs synthesis. Polymerization is the process in which small molecules combine to generate long-chain polymers. Polymerization is highly dependent on the concentration of the initiator and the monomer mixture. These values should be defined accordingly to avoid blockage problem in the microchannels, as they may directly impact the polymerization rate in the microreactor. In our study, we have considered the values used in the bulk process and adapted them to our micro-reactor system. While applying polymer solutions, there would be two major possibilities to hinder the efficiency of micro-reactor. (i) Clogging/blockage of channels in micro-reactor would occur since there would be possible smears of polymers on PDMS surface of channels. (ii) Though we did not observe in our process, polymer solutions would react to the microchannel surface. The critical point is that both solutions should simultaneously be introduced to the micro-reactor. This way, more uniform polymerization could be provided potentially. However, due to the different flow rates of both solutions (2.5 $\mu\text{L}/\text{min}$ and 143.75 $\mu\text{L}/\text{min}$), it was highly challenging to introduce them into the system at the same time. When the initiator was introduced into the micro-reactor before the monomer mixture, we observed some accumulations of initiator in the initial part of the main channel, which resulted in intense polymerization upon meeting the monomer mixture. The critical point is that both solutions should be introduced simultaneously to the micro-reactor. This way, more uniform polymerization could be provided potentially.

Durability and long-term stability are highly dependent on polymers utilized in MIP formation. HEMA-originated polymeric matrix used in this study have been tested for long-term durability and stability by us and the others in earlier studies ^{79,80}, and therefore, we did not perform such evaluations here. As an example, MIPs were challenged for 36 months of incubation at room temperature after they were coated on the sensors ⁸⁰. This long-term storage did not cause a statistically significant change in their performance, and the MIP-coated sensor response only

changed around 15%. Likewise, the researchers prepared synthetic cannabinoids imprinted polymers, and conducted stability analysis for 12 months, and there was no significant decrease in the sensor's performance⁷⁹. Moreover, lyophilization, often known as freeze-drying, can be used to increase durability of MIP nanoparticles⁸¹. On the other hand, long-term MIP synthesis was performed in the micro-reactor for 180 minutes, and we did not observe any performance issues for producing trillions of MIP nanoparticles on a single device. Evaluating the performance of micro-reactor for longer periods than 180 minutes would be beneficial for synthesizing more particles on a single batch. Last but not least, the proposed micro-reactor system stands out as a suitable alternative to the conventional counterparts for MIP synthesis for being cost-effective streamlined approach.” (Page 20-21)

REFERENCES

1. Ozcelikay, G. *et al.* Sensor-based MIP technologies for targeted metabolomics analysis. *TrAC Trends Anal. Chem.* **146**, 116487 (2022).
2. Inci, F., Celik, U., Turken, B., Özer, H. Ö. & Kok, F. N. Construction of P-glycoprotein incorporated tethered lipid bilayer membranes. *Biochem. Biophys. reports* **2**, 115–122 (2015).
3. Erdem, Ö. *et al.* Carbon-Based Nanomaterials and Sensing Tools for Wearable Health Monitoring Devices. *Adv. Mater. Technol.* **7**, 2100572 (2022).
4. Yucel, M. *et al.* Hand-Held Volatilome Analyzer Based on Elastically Deformable Nanofibers. *Anal. Chem.* **90**, 5122–5129 (2018).
5. Hui, Y. *et al.* Recent advancements in electrochemical biosensors for monitoring the water quality. *Biosensors* **12**, 551 (2022).
6. Ratautaite, V. *et al.* Evaluation of the interaction between SARS-CoV-2 spike glycoproteins and the molecularly imprinted polypyrrole. *Talanta* **253**, 123981 (2023).
7. Ratautaite, V. *et al.* Molecularly imprinted polypyrrole based sensor for the detection of SARS-CoV-2 spike glycoprotein. *Electrochim. Acta* **403**, 139581 (2022).
8. Antipchik, M., Reut, J., Ayankojo, A. G., Öpik, A. & Syritski, V. MIP-based electrochemical sensor for direct detection of hepatitis C virus via E2 envelope protein. *Talanta* **250**, 123737 (2022).
9. Jolly, P. *et al.* Aptamer–MIP hybrid receptor for highly sensitive electrochemical detection of prostate specific antigen. *Biosens. Bioelectron.* **75**, 188–195 (2016).
10. Balayan, S., Chauhan, N., Chandra, R. & Jain, U. Electrochemical based C-reactive protein (CRP) sensing through molecularly imprinted polymer (MIP) pore structure coupled with bi-metallic tuned screen-printed electrode. *Biointerface Res. Appl. Chem* **6**, 38 (2022).
11. Pacheco, J. G., Silva, M. S. V, Freitas, M., Nouws, H. P. A. & Delerue-Matos, C. Molecularly imprinted electrochemical sensor for the point-of-care detection of a breast cancer biomarker (CA 15-3). *Sensors Actuators B Chem.* **256**, 905–912 (2018).
12. Shafiee, H. *et al.* Acute on-chip hiv detection through label-free electrical sensing of

- viral nano-lysate. *Small* **9**, 2553–2563 (2013).
13. Samukaite-Bubniene, U. *et al.* Towards supercapacitors: Cyclic voltammetry and fast Fourier transform electrochemical impedance spectroscopy based evaluation of polypyrrole electrochemically deposited on the pencil graphite electrode. *Colloids Surfaces A Physicochem. Eng. Asp.* **610**, 125750 (2021).
 14. Zhao, Z., Yu, T., Miao, Y. & Zhao, X. Chloride ion-doped polyaniline/carbon nanotube nanocomposite materials as new cathodes for chloride ion battery. *Electrochim. Acta* **270**, 30–36 (2018).
 15. Wang, Y. *et al.* Urchin-like Ni_{1/3}Co_{2/3}(CO₃)_{0.5}·0.11 H₂O anchoring on polypyrrole nanotubes for supercapacitor electrodes. *Electrochim. Acta* **295**, 989–996 (2019).
 16. Ramanavicius, S. & Ramanavicius, A. Development of molecularly imprinted polymer based phase boundaries for sensors design (review). *Adv. Colloid Interface Sci.* **305**, 102693 (2022).
 17. Oztekin, Y., Ramanaviciene, A., Yazicigil, Z., Solak, A. O. & Ramanavicius, A. Direct electron transfer from glucose oxidase immobilized on polyphenanthroline-modified glassy carbon electrode. *Biosens. Bioelectron.* **26**, 2541–2546 (2011).
 18. Leonavicius, K., Ramanaviciene, A. & Ramanavicius, A. Polymerization model for hydrogen peroxide initiated synthesis of polypyrrole nanoparticles. *Langmuir* **27**, 10970–10976 (2011).
 19. Felix, F. S. & Angnes, L. Electrochemical immunosensors—a powerful tool for analytical applications. *Biosens. Bioelectron.* **102**, 470–478 (2018).
 20. Ramanavicius, A. *et al.* Synthesis of polypyrrole within the cell wall of yeast by redox-cycling of [Fe(CN)₆]³⁻/[Fe(CN)₆]⁴⁻. *Enzyme Microb. Technol.* **83**, 40–47 (2016).
 21. Apetrei, R.-M., Carac, G., Bahrim, G., Ramanaviciene, A. & Ramanavicius, A. Modification of *Aspergillus niger* by conducting polymer, Polypyrrole, and the evaluation of electrochemical properties of modified cells. *Bioelectrochemistry* **121**, 46–55 (2018).
 22. Apetrei, R.-M. *et al.* Cell-assisted synthesis of conducting polymer–polypyrrole–for the improvement of electric charge transfer through fungal cell wall. *Colloids Surfaces B Biointerfaces* **175**, 671–679 (2019).
 23. Ramanaviciene, A. & Ramanavicius, A. Molecularly imprinted polypyrrole-based synthetic receptor for direct detection of bovine leukemia virus glycoproteins. *Biosens. Bioelectron.* **20**, 1076–1082 (2004).
 24. Zeng, Q., Huang, X. & Ma, M. A molecularly imprinted electrochemical sensor based on polypyrrole/carbon nanotubes composite for the detection of S-ovalbumin in egg white. *Int. J. Electrochem. Sci.* **12**, 3965–3981 (2017).
 25. Devkota, L., Nguyen, L. T., Vu, T. T. & Piro, B. Electrochemical determination of tetracycline using AuNP-coated molecularly imprinted overoxidized polypyrrole

- sensing interface. *Electrochim. Acta* **270**, 535–542 (2018).
26. Wang, F., Zhu, L. & Zhang, J. Electrochemical sensor for levofloxacin based on molecularly imprinted polypyrrole–graphene–gold nanoparticles modified electrode. *Sensors Actuators B Chem.* **192**, 642–647 (2014).
 27. Ait-Touchente, Z. *et al.* High performance zinc oxide nanorod-doped ion imprinted polypyrrole for the selective electroensing of mercury II ions. *Appl. Sci.* **10**, 7010 (2020).
 28. Velepini, T., Pillay, K., Mbianda, X. Y. & Arotiba, O. A. Application of a polypyrrole/carboxy methyl cellulose ion imprinted polymer in the electrochemical detection of mercury in water. *Electroanalysis* **30**, 2612–2619 (2018).
 29. Tokel, O., Inci, F. & Demirci, U. Advances in plasmonic technologies for point of care applications. *Chem. Rev.* **114**, 5728–5752 (2014).
 30. Mataji-Kojouri, A., Ozen, M. O., Shahabadi, M., Inci, F. & Demirci, U. Entangled Nanoplasmonic Cavities for Estimating Thickness of Surface-Adsorbed Layers. *ACS Nano* **14**, 8518–8527 (2020).
 31. Inci, F. *et al.* Enhancing the nanoplasmonic signal by a nanoparticle sandwiching strategy to detect viruses. *Appl. Mater. Today* **20**, 100709 (2020).
 32. Parlatan, U. *et al.* Label-Free Identification of Exosomes using Raman Spectroscopy and Machine Learning. *Small* **19**, 2205519 (2023).
 33. Sadeque, M. S. Bin *et al.* Hydrogel-integrated optical fiber sensors and their applications: A comprehensive review. *J. Mater. Chem. C* (2023).
 34. Raziq, A. *et al.* Development of a portable MIP-based electrochemical sensor for detection of SARS-CoV-2 antigen. *Biosens. Bioelectron.* **178**, 113029 (2021).
 35. Bezdekova, J. *et al.* Magnetic molecularly imprinted polymers used for selective isolation and detection of *Staphylococcus aureus*. *Food Chem.* **321**, 126673 (2020).
 36. Pilvenyte, G. *et al.* Molecularly imprinted polymers for the determination of cancer biomarkers. *Int. J. Mol. Sci.* **24**, 4105 (2023).
 37. Pirzada, M., Sehit, E. & Altintas, Z. Cancer biomarker detection in human serum samples using nanoparticle decorated epitope-mediated hybrid MIP. *Biosens. Bioelectron.* **166**, 112464 (2020).
 38. Rebelo, T. S. C. R. *et al.* A Disposable Saliva Electrochemical MIP-Based Biosensor for Detection of the Stress Biomarker α -Amylase in Point-of-Care Applications. *Electrochem* **2**, 427–438 (2021).
 39. Battaglia, F. *et al.* Detection of canine and equine procalcitonin for sepsis diagnosis in veterinary clinic by the development of novel MIP-based SPR biosensors. *Talanta* **230**, 122347 (2021).
 40. Ben Hassine, A., Raouafi, N. & Moreira, F. T. C. Novel electrochemical molecularly

- imprinted polymer-based biosensor for Tau protein detection. *Chemosensors* **9**, 238 (2021).
41. Kou, Y. *et al.* Recyclable magnetic MIP-based SERS sensors for selective, sensitive, and reliable detection of paclitaxel residues in complex environments. *ACS Sustain. Chem. Eng.* **8**, 14549–14556 (2020).
 42. Sinha, A., Mugo, S. M., Lu, X. & Chen, J. Molecular imprinted polymer-based biosensors for the detection of pharmaceutical contaminants in the environment. in *Tools, Techniques and Protocols for Monitoring Environmental Contaminants* 371–389 (Elsevier, 2019).
 43. Urraca, J. L. *et al.* Polymeric complements to the Alzheimer's disease biomarker β -amyloid isoforms A β 1–40 and A β 1–42 for blood serum analysis under denaturing conditions. *J. Am. Chem. Soc.* **133**, 9220–9223 (2011).
 44. Kidakova, A. *et al.* Molecularly imprinted polymer-based SAW sensor for label-free detection of cerebral dopamine neurotrophic factor protein. *Sensors Actuators, B Chem.* **308**, 127708 (2020).
 45. Inci, F. *et al.* A disposable microfluidic-integrated hand-held plasmonic platform for protein detection. *Appl. Mater. Today* **18**, 100478 (2020).
 46. Saylan, Y. & Denizli, A. Molecular fingerprints of hemoglobin on a nanofilm chip. *Sensors (Switzerland)* **18**, (2018).
 47. Ansari, S. & Masoum, S. Molecularly imprinted polymers for capturing and sensing proteins: Current progress and future implications. *TrAC Trends Anal. Chem.* **114**, 29–47 (2019).
 48. He, Y. & Lin, Z. Recent advances in protein-imprinted polymers: synthesis, applications and challenges. *J. Mater. Chem. B* **10**, 6571–6589 (2022).
 49. Khumsap, T., Corpuz, A. & Nguyen, L. T. Epitope-imprinted polymers: applications in protein recognition and separation. *RSC Adv.* **11**, 11403–11414 (2021).
 50. Liu, Z. *et al.* A surface protein-imprinted biosensor based on boronate affinity for the detection of anti-human immunoglobulin G. *Microchim. Acta* **189**, 106 (2022).
 51. Ambrosini, S., Beyazit, S., Haupt, K. & Bui, B. T. S. Solid-phase synthesis of molecularly imprinted nanoparticles for protein recognition. *Chem. Commun.* **49**, 6746–6748 (2013).
 52. Sunayama, H. & Takeuchi, T. Protein-imprinted polymer films prepared via cavity-selective multi-step post-imprinting modifications for highly selective protein recognition. *Anal. Bioanal. Chem.* **413**, 6183–6189 (2021).
 53. Jahanban-Esfahlan, A., Roufegarinejad, L., Jahanban-Esfahlan, R., Tabibiazar, M. & Amarowicz, R. Latest developments in the detection and separation of bovine serum albumin using molecularly imprinted polymers. *Talanta* **207**, 120317 (2020).
 54. Perçin, I., Idil, N. & Denizli, A. Molecularly imprinted poly (N-isopropylacrylamide)

- thermosensitive based cryogel for immunoglobulin G purification. *Process Biochem.* **80**, 181–189 (2019).
55. Guo, Z., Zhao, J., He, H. & Liu, Z. Molecularly Imprinted and Cladded Nanotags Enable Specific SERS Bioimaging of Tyrosine Phosphorylation. *Chem. Asian J.* **17**, e202200844 (2022).
 56. Lin, C.-C. *et al.* Embedded upconversion nanoparticles in magnetic molecularly imprinted polymers for photodynamic therapy of hepatocellular carcinoma. *Biomedicines* **9**, 1923 (2021).
 57. Cantarella, M. *et al.* Molecularly imprinted polymer for selective adsorption of diclofenac from contaminated water. *Chem. Eng. J.* **367**, 180–188 (2019).
 58. Hua, M. Z., Feng, S., Wang, S. & Lu, X. Rapid detection and quantification of 2, 4-dichlorophenoxyacetic acid in milk using molecularly imprinted polymers–surface-enhanced Raman spectroscopy. *Food Chem.* **258**, 254–259 (2018).
 59. Sorribes-Soriano, A. *et al.* Magnetic molecularly imprinted polymers for the selective determination of cocaine by ion mobility spectrometry. *J. Chromatogr. A* **1545**, 22–31 (2018).
 60. Zeng, H., Yu, X., Wan, J. & Cao, X. Rational design and synthesis of molecularly imprinted polymers (MIP) for purifying tylosin by seeded precipitation polymerization. *Process Biochem.* **94**, 329–339 (2020).
 61. Phungpanya, C. *et al.* Synthesis of prednisolone molecularly imprinted polymer nanoparticles by precipitation polymerization. *Polym. Adv. Technol.* **29**, 3075–3084 (2018).
 62. Wang, Z. *et al.* Facile fabrication of snowman-like magnetic molecularly imprinted polymer microspheres for bisphenol A via one-step Pickering emulsion polymerization. *React. Funct. Polym.* **164**, 104911 (2021).
 63. Decompte, E., Lobaz, V., Monperrus, M., Deniau, E. & Save, M. Molecularly imprinted polymer colloids synthesized by miniemulsion polymerization for recognition and separation of nonylphenol. *ACS Appl. Polym. Mater.* **2**, 3543–3556 (2020).
 64. Liu, X. *et al.* Synthesis of molecularly imprinted polymer by suspension polymerization for selective extraction of p-hydroxybenzoic acid from water. *J. Appl. Polym. Sci.* **136**, 46984 (2019).
 65. Zhang, W., Li, Q., Cong, J., Wei, B. & Wang, S. Mechanism analysis of selective adsorption and specific recognition by molecularly imprinted polymers of Ginsenoside Re. *Polymers (Basel)*. **10**, 216 (2018).
 66. Banan, K. *et al.* Nano-sized magnetic core-shell and bulk molecularly imprinted polymers for selective extraction of amiodarone from human plasma. *Anal. Chim. Acta* **1198**, 339548 (2022).
 67. Qiu, L. *et al.* Core–shell molecularly imprinted polymers on magnetic yeast for the

- removal of sulfamethoxazole from water. *Polymers (Basel)*. **12**, 1385 (2020).
68. Ramanavičius, S. *et al.* Electrochemically deposited molecularly imprinted polymer-based sensors. *Sensors* **22**, 1282 (2022).
 69. Akgönüllü, S., Kılıç, S., Esen, C. & Denizli, A. Molecularly imprinted polymer-based sensors for protein detection. *Polymers (Basel)*. **15**, 629 (2023).
 70. Yarman, A., Kurbanoglu, S., Zebger, I. & Scheller, F. W. Simple and robust: The claims of protein sensing by molecularly imprinted polymers. *Sensors Actuators B Chem.* **330**, 129369 (2021).
 71. Parisi, O. I. *et al.* The evolution of molecular recognition: From antibodies to molecularly imprinted polymers (MIPs) as artificial counterpart. *J. Funct. Biomater.* **13**, 12 (2022).
 72. Pulingam, T., Foroozandeh, P., Chuah, J.-A. & Sudesh, K. Exploring various techniques for the chemical and biological synthesis of polymeric nanoparticles. *Nanomaterials* **12**, 576 (2022).
 73. He, J.-X., Pan, H.-Y., Xu, L. & Tang, R.-Y. Application of molecularly imprinted polymers for the separation and detection of aflatoxin. *J. Chem. Res.* **45**, 400–410 (2021).
 74. Kadhem, A. J., Gentile, G. J. & Fidalgo de Cortalezzi, M. M. Molecularly imprinted polymers (MIPs) in sensors for environmental and biomedical applications: a review. *Molecules* **26**, 6233 (2021).
 75. Rahman, S. *et al.* Molecularly imprinted polymers (MIPs) combined with nanomaterials as electrochemical sensing applications for environmental pollutants. *Trends Environ. Anal. Chem.* e00176 (2022).
 76. Prasad, B. B. & Pathak, P. K. Development of surface imprinted nanospheres using the inverse suspension polymerization method for electrochemical ultra sensing of dacarbazine. *Anal. Chim. Acta* **974**, 75–86 (2017).
 77. Orbay, S., Kocaturk, O., Sanyal, R. & Sanyal, A. Molecularly imprinted polymer-coated inorganic nanoparticles: Fabrication and biomedical applications. *Micromachines* **13**, 1464 (2022).
 78. Niu, M., Pham-Huy, C. & He, H. Core-shell nanoparticles coated with molecularly imprinted polymers: a review. *Microchim. Acta* **183**, 2677–2695 (2016).
 79. Akgönüllü, S., Battal, D., Yalcin, M. S., Yavuz, H. & Denizli, A. Rapid and sensitive detection of synthetic cannabinoids JWH-018, JWH-073 and their metabolites using molecularly imprinted polymer-coated QCM nanosensor in artificial saliva. *Microchem. J.* **153**, 104454 (2020).
 80. Gerdan, Z., Saylan, Y., Uğur, M. & Denizli, A. Ion-imprinted polymer-on-a-sensor for copper detection. *Biosensors* **12**, 91 (2022).
 81. Safaryan, A. H. M. *et al.* Optimisation of the preservation conditions for molecularly

imprinted polymer nanoparticles specific for trypsin. *Nanoscale Adv.* **1**, 3709–3714 (2019).